# Ten months of temporal variation in the clinical journey of hospitalised patients with COVID-19: An observational cohort

ISARIC Clinical Characterisation Group, Matthew D Hall[1]*, Joaquín Baruch[2], Gail Carson[2], Barbara Wanjiru Citarella[2], Andrew Dagens[2], Emmanuelle A Dankwa[3], Christl A Donnelly[3,4], Jake Dunning[2], Martina Escher[2], Christiana Kartsonaki[5], Laura Merson[2,6], Mark Pritchard[2], Jia Wei[1], Peter W Horby[2], Amanda Rojek[2,7], Piero L Olliaro[2]

[1]Big Data Institute, Nuffield Department of Medicine, University of Oxford, Oxford, United Kingdom; [2]ISARIC Global Support Centre, Centre for Tropical Medicine and Global Health, Nuffield Department of Medicine, University of Oxford, Oxford, United Kingdom; [3]Department of Statistics, University of Oxford, Oxford, United Kingdom; [4]MRC Centre for Global Infectious Disease Analysis, Abdul Latif Jameel Institute for Disease and Emergency Analytics and Department of Infectious Disease Epidemiology, Imperial College London, London, United Kingdom; [5]MRC Population Health Research Unit, Clinical Trials Service Unit and Epidemiological Studies Unit, Nuffield Department of Population Health, University of Oxford, Oxford, United Kingdom; [6]Infectious Diseases Data Observatory, Centre for Tropical Medicine and Global Health, University of Oxford, Oxford, United Kingdom; [7]Royal Melbourne Hospital, Melbourne, Australia Centre for Integrated Critical Care, University of Melbourne, Melbourne, Australia

*For correspondence: matthew.hall@bdi.ox.ac.uk

Group author details: ISARIC Clinical Characterisation Group See page 22

Competing interest: The authors declare that no competing interests exist.

## Abstract

**Background:** There is potentially considerable variation in the nature and duration of the care provided to hospitalised patients during an infectious disease epidemic or pandemic. Improvements in care and clinician confidence may shorten the time spent as an inpatient, or the need for admission to an intensive care unit (ICU) or high dependency unit (HDU). On the other hand, limited resources at times of high demand may lead to rationing. Nevertheless, these variables may be used as static proxies for disease severity, as outcome measures for trials, and to inform planning and logistics.

**Methods:** We investigate these time trends in an extremely large international cohort of 142,540 patients hospitalised with COVID-19. Investigated are: time from symptom onset to hospital admission, probability of ICU/HDU admission, time from hospital admission to ICU/HDU admission, hospital case fatality ratio (hCFR) and total length of hospital stay.

**Results:** Time from onset to admission showed a rapid decline during the first months of the pandemic followed by peaks during August/September and December 2020. ICU/HDU admission was more frequent from June to August. The hCFR was lowest from June to August. Raw numbers for overall hospital stay showed little variation, but there is clear decline in time to discharge for ICU/HDU survivors.

**Conclusions:** Our results establish that variables of these kinds have limitations when used as outcome measures in a rapidly evolving situation.

**Funding:** This work was supported by the UK Foreign, Commonwealth and Development Office and Wellcome [215091/Z/18/Z] and the Bill & Melinda Gates Foundation [OPP1209135]. The funders

had no role in study design, data collection and analysis, decision to publish, or preparation of the manuscript.

## Editor's evaluation

This large multicenter study tracked the clinical journeys of COVID-19 hospitalized patients over 2020, and found variations in clinical outcomes over time. This paper will be of interest to the large class of clinicians, public health workers and health policy makers who want to know the variation in the nature and duration of the care provided to hospitalised patients during an infectious disease epidemic. The study highlights the importance of maintaining the capacity of registration of infectious disease like COVID-19, during a pandemic and after. While the cohort recruited patients from multiple countries, the vast majority of patients came from the UK, so the results are most applicable to this country.

## Introduction

During an epidemic or pandemic of a novel infectious disease, variations in the duration of each stage of a hospitalised patient's progress from symptom onset, to hospital admission, and hence to outcome are critical for an effective response. Clinicians use these data as a proxy for disease severity, and to provide prognostic information to patients and their families. Policy makers use these data to inform system wide planning for staffing, infrastructure, to predict requirements for consumables (such as personal protective equipment), and to assess performance of the hospital system. And for clinical research, these measures are used as trial outcomes to determine the efficacy of novel treatments.

Often, the extent to which patient journeys vary during an epidemic is not understood. There are changes in clinical practice (*World Health Organisation, 2021*) – clinical understanding of the natural history of diseases improves with time (*Docherty et al., 2021*), and so too does confidence in safe discharge criteria or in alternative models of care (*Rojek and Horby, 2016*), such as remote monitoring (*Nunan et al., 2020*; *Bell et al., 2021*). Moreover, the introduction of effective treatments (*Rochwerg et al., 2020*) and standardisation of care may rapidly reduce the severity or time course of illness (*Dennis et al., 2021*). However, decisions about whether to admit or escalate care are also dependent on logistic factors such as the availability of resources (e.g. ventilators, intensive care beds, staff) that may be rationed during the peak of a pandemic, but abundant at other phases of an outbreak (*Tyrrell et al., 2021*; *National Institute for Health and Care Excellence, 2021*; *Pagel et al., 2020*). There may also be changes in policy to admit patients for indications that are not clinical – such as to facilitate effective quarantine (*Wuhan Novel Coronavirus, 2021*) or supervise provision of treatments in clinical trials. We hypothesise that there is significant variation in the patient journey over a pandemic period, and that this variability may limit the way these data can be responsibly used.

In this paper, we assess temporal changes in hospital admission, length of stay, and escalation of care for hospitalised patients with SARS-CoV-2 infection included in the International Severe Acute Respiratory and emerging Infection Consortium (ISARIC) WHO Clinical Characterisation Protocol International cohort (*ISARIC Clinical Characterisation Group, 2020*). This is to our knowledge the largest, prospective international cohort including standardised clinical data, and, as of the time of writing, includes data collected from 26 January 2020 to 20 September 2021 on 708,085 people hospitalised with COVID-19 in 1669 sites in 64 countries.

We use this dataset to determine whether these variables did indeed change over the course of the SARS-CoV-2 pandemic during 2020, and where there are changes, explore if there are predictable influences that account for this.

## Methods

As previously described (*ISARIC Clinical Characterisation Group, 2021*), eligible for recruitment were patients with confirmed or suspected COVID-19 infection admitted to an ISARIC partner site and submitted to the ISARIC-hosted REDCap system. Additional data contributed to ISARIC via other mechanisms have not been included due to differences in data structure. The datasets used in analyses in this paper are drawn from a population of all patients with a symptom onset date, or

hospital admission date, recorded between March and December 2020 inclusive. Follow-up could be conducted until 8 March 2021, at which point the dataset was closed. Some patients had a recorded hospital admission date before their symptom onset date. In most cases, this would represent a nosocomial infection, but sometimes it could instead be that the patient was coincidentally admitted for a separate medical condition during their incubation period. In either case their admission date would not represent the start of their hospital treatment for COVID-19, and so we recorded the latter variable, hereafter 'COVID-19 admission date', as the later of the symptom onset date and hospital admission date. Patients were followed until they left the study site, due to death, hospital discharge, or another reason (such as transfer to another facility). Patients lost to follow-up before any of these outcome events were included, unless the time to that event was the variable of interest in a particular analysis. (For example, in question 5, below, time to death and discharge were used as dependent variables, and patients lost to follow-up, for whom this time was not recorded, were excluded.)

We used the complete dataset described above to explore temporal variation in six variables or collections of variables. Not all patients had recorded information for the variables of interest in each question, so, in each case, a subset was analysed. The questions and data subsets were as follows:

**Question 1**

Variation in the time from symptom onset to hospital admission. Patients were excluded if this variable was not available, or if they were admitted prior to symptom onset.

**Question 2**

Variation in the proportion of patients being admitted to an ICU or HDU. Patients were excluded if this variable was not available.

**Question 3**

Variation in the time from COVID-19 admission to ICU/HDU admission. Patients were excluded if they were never admitted to an ICU or HDU, or if this variable was otherwise not available.

**Question 4**

Variation in the overall case fatality rate. Patients were excluded if the final outcome of their hospital stay was either not recorded or recorded as something other than "death" or "discharge" (for example, transfer to another facility).

**Question 5**

Variation in the time from COVID-19 admission to death or discharge. (We describe either as an "outcome".) Exclusions were as in question 4, as well as patients who had a recorded outcome but no recorded outcome date.

**Question 6**

Variation in the status of patients (admitted, ICU/HDU admitted, dead, discharged, or unknown outcome) on a given day after admission. Excluded here were patients whose ICU/HDU status on the day of admission was unknown.

Further filtering was done to (a) remove any nonsensical values (such as recorded time of hospital admission after hospital exit), (b) remove patients admitted to hospital in 2021 for all questions except 1 (such patients were included when exploring the latter due to right censoring concerns if they were omitted), and (c) when considering hospital admission (question 1), ICU/HDU admission (questions 2 and 3) or final outcome (questions 4–6), exclude patients for whom the time to that event was in the top 2.5 % of recorded values (as this range included extreme outliers that may have been the result of incorrect data entry). We designate these three limits by $l_{admission}$, $l_{ICU}$ and $l_{outcome}$ respectively.

The values of $l_{ICU}$ and $l_{outcome}$ also define a time period over which the events of ICU/HDU admission and final outcome (respectively) can be defined for questions 2 and 4. Thus, in question 4, the actual variable of interest is death or discharge within $l_{outcome}$ days. For question 2, the variable is ICU/HDU admission within $l_{ICU}$ days of observation. As a result, we additionally excluded patients with incomplete follow-up who were observed for less than $l_{ICU}$ days from COVID-19 admission without experiencing ICU/HDU admission, as such an event may have occurred within the time limit without it being observed.

For all analyses with a single outcome variable, we plotted its mean value against the epidemiological week of symptom onset (question 1) or COVID-19 admission (others), both overall and with respect to various variables of interest (e.g. age group).

For the exploration of patient status by day after COVID-19 admission (question 6), the progress of a patient along the course of their hospital stay was visualised by means of Sankey diagrams. Five

states were considered: ward occupancy, ICU/HDU occupancy, the final outcomes of death or discharge, and unknown outcome. We recorded the state of each patient on the day of admission, on every subsequent day, and their final outcome. Where a patient's exact location (ward or ICU/HDU) in the hospital was not recorded on a given day, their last known location was used. For the figures in this article, we present only the data on day of COVID-19 admission (A), 3 days later (A + 3), 7 days later (A + 7), and final outcome (O + 1). An interactive version of this diagram is currently under development and will be made available to the research community as soon as possible.

The four most frequent symptoms at admission were cough, fatigue, fever, and shortness of breath. We introduced a new variable to the dataset counting the number of these present at admission for each patient. Missing data was disregarded here, so this represents a lower bound.

## Statistical analysis

Multivariable linear regression was used to investigate factors associated with time from onset of symptoms to hospital admission (question 1), time from COVID-19 admission to ICU/HDU admission (question 3), and time from COVID-19 admission to death or discharge (question 5). In all cases, the dependent variable was log-transformed and a pseudocount of 1 added in order to prevent taking logarithms of zero. Multivariable logistic regression was used to investigate factors associated with ICU/HDU admission (question 2) and fatal outcome (question 5), and to adjust for these factors as potential confounders for our primary outcome variables; for a full list of these variables, see *Supplementary file 1*. For all regression analyses, we analysed the presence of comorbidities as covariables. As there was a considerable amount of missing data for each of these, we introduced an 'unknown' class to the regression models for these variables rather than exclude patients without values for them entirely. After this modification, every regression analysis was performed as a complete case analysis.

The significance of every dependent variable in every model, including interaction terms used in regressions for hCFR (question 4) and time to outcome (question 5) was assessed using the Wald test.

## Software

All analyses were performed in R 4.0.3 (*R Development Core Team, 2013*), with packages including the *tidyverse* (*Wickham et al., 2019*), and *ggalluvial* (*Brunson, 2020*). Code for processing the data and performing the regression analyses is available, copy archived at swh:1:rev:ce42035d6cf80852089d-95264215f7bb487cb998 (*Hall, 2021*).

**Table 1.** Baseline characteristics of the included patients.

*Some patients admitted in early 2021 are included in order to fully represent patients with symptom onset in December 2020.

| Variable | Value | Count | % |
|---|---|---|---|
| Month of admission | March | 27,108 | 19.4 |
| | April | 42,267 | 30.3 |
| | May | 12,311 | 8.82 |
| | June | 5,342 | 3.83 |
| | July | 2,811 | 2.01 |
| | August | 2,218 | 1.59 |
| | September | 5,265 | 3.77 |
| | October | 13,822 | 9.91 |
| | November | 15,155 | 10.9 |
| | December | 13,205 | 9.47 |
| Sex | Female | 59,719 | 42.8 |
| | Male | 79,550 | 57 |
| | Unknown | 235 | 0.168 |
| Age group | 0–19 | 2,697 | 1.93 |
| | 20–39 | 9,302 | 6.67 |
| | 40–59 | 30,399 | 21.8 |
| | 60–69 | 22,815 | 16.4 |
| | 70–79 | 29,901 | 21.4 |
| | 80+ | 41,571 | 29.8 |
| | Unknown | 2,819 | 2.02 |
| Symptom onset post-admission | No | 118,874 | 85.2 |
| | Yes | 11,695 | 8.38 |
| | Unknown | 8,935 | 6.4 |

# Results

## Patient characteristics

Our complete dataset consisted of 142,540 patients (60,977 female, 81,325 male, 238 unknown sex), of median age 70 [IQR 56–82], admitted at 620 sites in 47 countries. *Table 1* shows a summary of baseline characteristics, and more detail, including country of origin and cross tabulation by month of admission, can be found in *Supplementary file 2*.

*Table 2* shows the prevalence of symptoms at admission and comorbidities. A total of 1030 individuals (0.7%) were pregnant women.

A basic summary of the various components of the patient journey that we investigated can be found in *Table 3*.

## Time from symptom onset to hospital admission (Question 1)

A total of 11,695 patients (8.2%) were recorded as having symptom onset following hospital admission, while for 8,935 (6.3%) patients this information was missing. After excluding these, we analysed length of illness before hospitalisation for those patients for whom it was recorded (n = 127,915, 89.7%). The 97.5 % quantile of time to admission ($l_{admission}$) was 24 days, and patients with recorded durations longer than this were excluded as described above. The median time from symptom onset to admission was 5 days (IQR 1–8). This variable showed a marked decline during March, from a median of 9 days (IQR 5–14) for patients with onset in the week beginning March 1–3 (IQR 0–7) in that beginning April 5. Little further variation occurred until late July, when a gradual increase started, which then peaked at a median of 6 (IQR 2–9) for the weeks in late August and early September before a decline to a low of a median 4 (IQR 1–7) days in November; this was followed by another slight increase in December. Times from onset of symptoms to admission were shortest in the oldest and youngest age groups (*Figure 1b*). Patients with a fatal outcome had, generally, shorter time from onset of symptoms until admission compared to survivors (*Figure 1c*).

The four most frequent symptoms at admission were cough, fever, shortness of breath, and fatigue; we class these as 'common' (see Methods). The number of these that were present increased with time to hospital admission (*Figure 1d*), with the shortest durations of all occurring amongst those presenting with none of them. Amongst the 4636 patients in this analysis presenting with none, the most common other symptoms were confusion (51.6%), vomiting (31.7%), abdominal pain (26%), and diarrhoea (18.9%). Within this group, confusion was the single most common presenting symptom documented in patients over 60, while in younger age groups the most prevalent symptoms were gastrointestinal (see *Supplementary file 3*).

We further explored this question using a multivariable linear regression analysis (*Supplementary file 4*). When compared to April, times from symptom onset to admission were shorter in May and June, and longer in March and from August onwards (all p < 0.01; see *Supplementary file 4* for confidence intervals). There was very strong evidence of an overall effect of month of onset on time from onset to admission (Wald test p < 0.001). Patients aged 40–59 showed the longest times to admission when compared to any other age group (all p < 0.001; see *Supplementary file 4* for CIs). Time from symptom onset to admission was also positively associated with the number of 'common' symptoms (25.1 % increase per symptom, 95% CI 24.5%–25.7%), male sex (3.3 % increase, 95% CI 2.2%–4.4%) and discharge as the final outcome (14.3 % increase, 95% CI 12.9%–15.7%).

## ICU/HDU admission and time to ICU/HDU admission (questions 2 and 3)

Of the 139,504 patients with COVID-19 admission in 2020, 136,849 (98.1%) had recorded data on whether they ever admitted to an ICU/HDU or not; of these, 28,171 (20.6%) had been admitted at least once. Where time to ICU/HDU was recorded, the 97.5 % quantile of this duration ($l_{ICU}$) was 13 days. We excluded patients for whom this variable was greater than that or unknown, along with those whose outcome was unknown and who had follow-up for less than 13 days with no ICU/HDU admission, for a total of 122,368 patients. The outcome variable in this section is thus ICU/HDU admission within 13 days of COVID-19 admission. The proportion of individuals experiencing this showed a marked decline over March followed by a renewed peak, and then subsequent decline, in June through August (*Figure 2A*). The oldest age group (80+) had by far the smallest proportion

**Table 2.** Prevalence of symptoms at hospital admission and comorbidities.

The final column gives the number of times the condition is recorded as present over the number of times its presence or absence is recorded (i.e. the data is non-missing). Designated "common" symptoms are indicated with a (C); the number and percentages of patients presenting with combinations of these are separately presented.

| | Name | % present | N (present)/n (data recorded) |
|---|---|---|---|
| Symptoms at admission | Cough (C) | 66.6 | (87218/131002) |
| | Shortness of breath (C) | 64.4 | (89611/139244) |
| | Fever (C) | 63.4 | (84665/133494) |
| | Fatigue (C) | 44.7 | (52837/118184) |
| | Confusion | 24.9 | (31167/125123) |
| | Vomiting | 19.9 | (24577/123625) |
| | Myalgia | 18.8 | (20921/111419) |
| | Diarrhoea | 18.2 | (22375/123121) |
| | Headache | 12 | (13424/112069) |
| | Abdominal pain | 11.1 | (13294/120175) |
| | Ageusia | 8.8 | (6758/76396) |
| | Wheezing | 7.7 | (8846/115511) |
| | Anosmia | 6.8 | (5281/77751) |
| | Runny nose | 3.4 | (3704/108623) |
| | Ulcers | 2.2 | (2291/105394) |
| | Bleeding | 1.8 | (2093/119266) |
| | Rash | 1.5 | (1713/113636) |
| | Seizures | 1.5 | (1801/120755) |
| | Lymphadenopathy | 0.7 | (774/112245) |
| | Conjunctivitis | 0.5 | (553/113083) |
| | Ear pain | 0.5 | (484/94873) |
| Number of recorded 'common' symptoms (C) | 0 | 7.6 | (10836/142540) |
| | 1 | 20.5 | (29257/142540) |
| | 2 | 26.4 | (37681/142540) |
| | 3 | 29 | (41359/142540) |
| | 4 | 16.4 | (23407/142540) |
| Comorbidities | Hypertension | 47.6 | (50174/105433) |
| | Chronic cardiac disease | 29.7 | (38175/128374) |
| | Diabetes | 16.8 | (20037/119155) |
| | Chronic pulmonary disease | 16.5 | (22040/133662) |
| | Chronic kidney disease | 15.7 | (20894/133256) |
| | Obesity | 14.4 | (16624/115463) |
| | Asthma | 13.2 | (17656/133341) |
| | Dementia | 12.9 | (16404/127239) |
| | Smoking | 12.8 | (7299/57164) |

*Table 2 continued on next page*

*Table 2 continued*

| | Name | % present | N (present)/n (data recorded) |
|---|---|---|---|
| | Chronic neurological disorder | 11.5 | (15248/132789) |
| | Rheumatological disorder | 11.2 | (13814/123453) |
| | Malignant neoplasm | 9.3 | (12343/132537) |
| | Chronic haemotologic disease | 4.1 | (5117/123739) |
| | Liver disease | 3.5 | (4443/128733) |
| | Malnutrition | 2.6 | (3094/119518) |
| | HIV/AIDS | 0.4 | (515/119235) |

of ICU/HDU admissions over the whole timeline (5.2%, compared with, for example, 33.5 % in the age-group 60–69). In a multivariable logistic regression model (*Supplementary file 5*), the following patterns were observed: there were higher odds of ICU/HDU admission during all months except May and November, when compared to April (all p < 0.05, see *Supplementary file 5* for CIs); those aged 80+ had lower odds of ICU/HDU admission (OR 0.12 for admission when compared to the 40–59 age group, 95% CI 0.11–0.13). Males were more likely to be admitted to ICU/HDU (OR 1.57, 95% CI 1.45–1.63). Patients who died had greatly increased odds of having been previously admitted (OR 6.1, 95% CI 5.8–6.41). Compared to those with symptom onset less than a week before hospital admission, patients with admitted prior to onset had lower odds of being admitted to ICU/HDU (OR 0.32, 95% CI 0.28–0.36), whereas those with longer times to hospital admission had increased odds (OR 1.43 for 7–13 days, 95% CI 1.37–1.5, 1.31 for 14 or more days, 95% CI 1.22–1.4). An overall effect of month of COVID admission on odds of ICU/HDU admission was highly significant (Wald test p < 0.0001). Comorbidities associated with higher odds of admission were hypertension (OR 1.27, 95% CI 1.21–1.34) and obesity (OR 1.78, 95% CI 1.69–1.88), whereas a wide variety of serious or chronic medical conditions were associated with lower odds (see *Supplementary file 5*), as was smoking (OR 0.79, 95% CI 0.72–0.87). The most extreme fitted odds ratio for a comorbidity with a positive association was 1.78 for obesity, while that for an inverse association was 0.19 for dementia.

Of the 28,171 patients with recorded ICU/HDU admission, 27,167 (96.4%) had non-missing data for time from COVID-19 admission to first ICU/HDU admission. The 97.5 % quantile rule again excluded patients whose value of this variable was greater than 13 days. The median time to ICU/HDU was 1 day (IQR 0–2). Raw time trends in this variable were modest (*Figure 2B*). Multivariable linear regression (*Supplementary file 6*) nevertheless did show evidence for an overall association with month of COVID-19 admission (Wald test p < 0.001), with, when compared to April, evidence for longer times to ICU/HDU in March, October, November and December (all p < 0.05; see *Supplementary file 6*). Time to ICU/HDU also showed a general increase with age (Wald test for overall association p < 0.001). There was no evidence of an association with final outcome (death or discharge) or with sex. Compared to patients admitted to hospital within a week of symptom onset, those admitted prior to onset had a 67.6 % increase in time to ICU/HDU (95% CI 55.9–80.2%) while those with longer times to admission had shorter times to ICU/HDU (8.6 % decrease for 7–13 days, 95% CI 6.6–10.6%, 14.9 % decrease for 14 or more days, 95% CI 11.9–17.8%). Comorbidities associated with longer time to ICU/HDU were asthma (6.5 % increase, 95% CI 3.3–9.7%), chronic haematological disease (20.4 % increase, 95% CI 12.6–28.8%), and chronic kidney disease (9.9 % increase, 95% CI 6–14.1%). In contrast, obesity (7.7 % decrease, 95% CI 5.2–10%), diabetes (3.7 % decrease, 95% CI 0.9–6.3%) and smoking (5.2 % decrease, 95% CI 0.4–9.8%) were associated with shorter time to ICU. There was also evidence of a longer time to ICU/HDU amongst pregnant patients (compared to non-pregnant females) (15.6%, 95% CI 1.7–31.4%).

**Table 3.** Summary of the components of the inpatient journey and their variation over the course of 2020. All time periods are in days. Patients are categorised by month of symptom onset for onset to admission, and by month of COVID admission in all other cases. Patients with COVID admission in 2021, who are included in the analysis of time from onset to admission if their onset date was in 2020, are not listed here as they are excluded from any analysis where the outcome variable is not time from onset to admission. "Outcome" is either death or discharge, and the 'admission to outcome' column gives the total length of hospital stay. For all durations, the top 2.5 % of values are excluded as potentially mis-entered.

| Month | Onset to hospital admission | | Proportion entering ICU/HDU | COVID-19 admission to ICU/HDU | | hCFR | COVID-19 admission to death | | COVID-19 admission to discharge | | COVID-19 admission to outcome | |
|---|---|---|---|---|---|---|---|---|---|---|---|---|
| | Mean | SD | | Mean | SD | | Mean | SD | Mean | SD | Mean | SD |
| March | 6.82 | 5.15 | 0.25 | 1.78 | 2.34 | 0.33 | 11 | 8.41 | 10.9 | 9.71 | 10.8 | 9.29 |
| April | 4.27 | 4.53 | 0.16 | 1.63 | 2.36 | 0.33 | 9.1 | 7.96 | 10.3 | 9.14 | 9.91 | 8.78 |
| May | 4.09 | 4.58 | 0.17 | 1.45 | 2.45 | 0.3 | 10 | 8.41 | 11 | 9.62 | 10.9 | 9.28 |
| June | 4.4 | 4.51 | 0.3 | 1.04 | 2.12 | 0.27 | 10 | 8.65 | 10.5 | 8.83 | 10.5 | 8.78 |
| July | 4.77 | 4.22 | 0.35 | 1.1 | 2.29 | 0.21 | 11 | 8.68 | 9.53 | 8.37 | 9.88 | 8.46 |
| August | 5.49 | 4.6 | 0.36 | 1.41 | 2.49 | 0.22 | 12 | 8.82 | 8.88 | 7.88 | 9.44 | 8.16 |
| September | 6.3 | 5.01 | 0.24 | 1.38 | 2.21 | 0.22 | 14 | 9.81 | 9.25 | 8.64 | 10.2 | 9.08 |
| October | 5.72 | 4.89 | 0.19 | 1.69 | 2.59 | 0.26 | 12 | 8.69 | 9.78 | 8.75 | 10.5 | 8.81 |
| November | 5.17 | 4.75 | 0.18 | 1.48 | 2.49 | 0.26 | 12 | 8.38 | 9.1 | 8.08 | 9.78 | 8.24 |
| December | 4.42 | 4.21 | 0.22 | 1.51 | 2.48 | 0.29 | 11 | 8 | 9.52 | 8.16 | 10 | 8.15 |

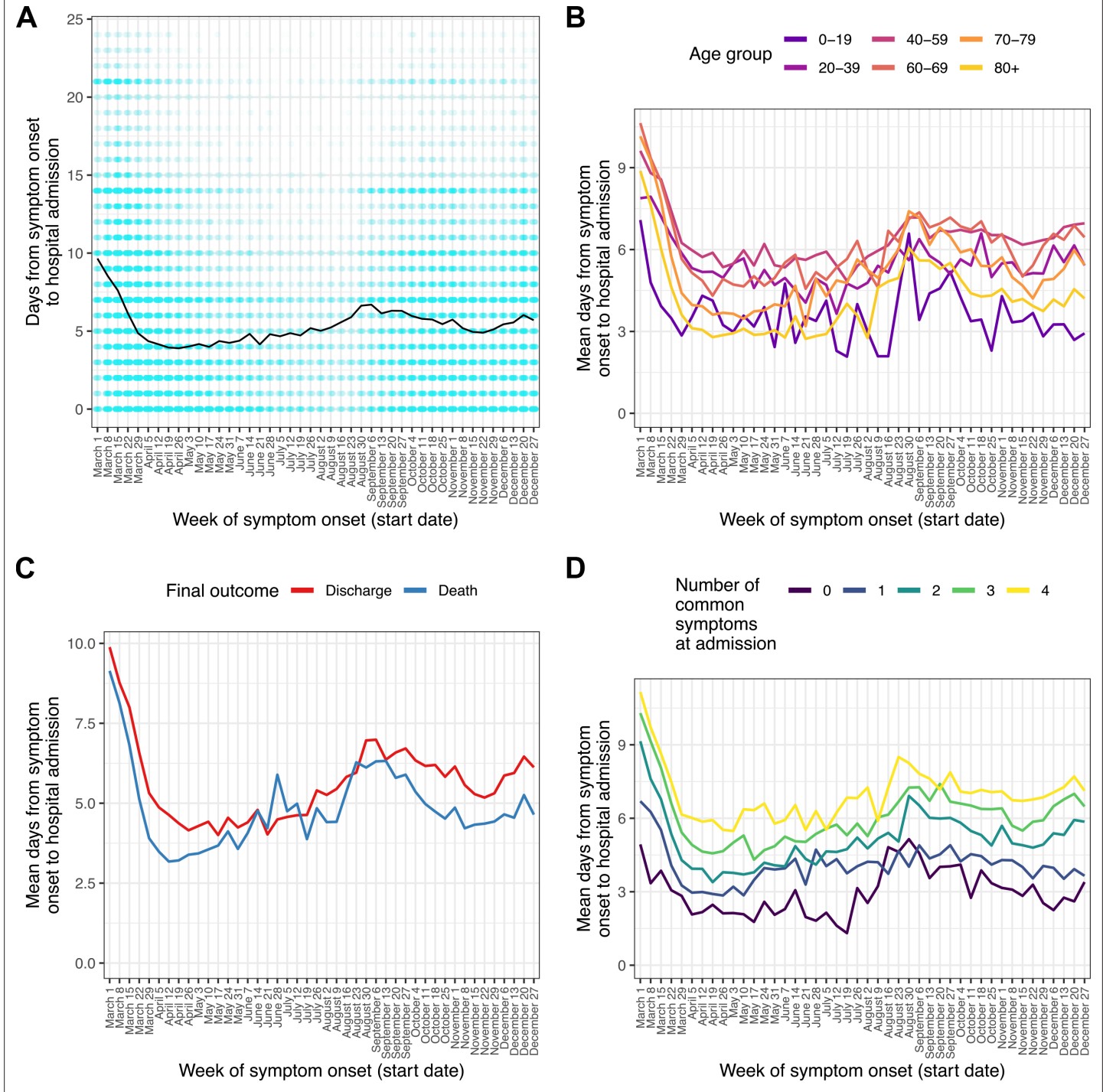

**Figure 1.** Time from reported symptom onset to hospital admission, by week of reported symptom onset. (**A**) Blue cells represent binned patients, with darker colours corresponding to more individuals. The black line represents the mean. (**B**)-(**D**) Mean time to admission plotted by patient characteristics: (**B**) age group, (**C**) final outcome, (**D**) number of the four most common symptoms (cough, fatigue, fever, and shortness of breath) present upon admission.

The online version of this article includes the following source data for figure 1:

**Source data 1.** Number of individuals for each combination of week of symptom onset and count of days from symptom onset to admission.

**Source data 2.** Mean number of days from symptom onset to admission by week of symptom onset.

**Source data 3.** Mean number of days from symptom onset to admission by week of symptom onset, by age group.

*Figure 1 continued on next page*

*Figure 1 continued*

**Source data 4.** Mean number of days from symptom onset to admission by week of symptom onset, by final outcome (death or discharge).

**Source data 5.** Mean number of days from symptom onset to admission by week of symptom onset, by number of common symptoms recorded at admission.

## Case fatality rate and time from COVID-19 admission to outcome (questions 4 and 5)

We next analysed the final outcome of death or discharge, and the total time from hospital admission to one of those outcomes, in a set of 116,537 patients admitted during 2020 with one of those outcomes recorded (83.5 % of the total admitted during 2020). The 97.5 % quantile of time to outcome ($l_{outcome}$) was 45 days, and once more patients with recorded durations in hospital longer than this were excluded; as a result, in practice the outcome here is death or discharge within 45 days. (As patient data was collected until 8 March 2021 and all patients here were admitted in 2020, patients admitted at the end of the period of interest had the same chance of complete follow-up as any other.) The raw hCFR was 0.3. The median time to death was 8 days (IQR 4–15) and to discharge 7 (IQR 4–14). (Amongst patients with no recorded outcome, excluded here, the median follow-up time was 9 days with an IQR of 2–22; the median follow-up for all patients regardless of outcome, recorded or not, was also 9 days with an IQR of 5–16). Over the entire 10-month period of interest (*Figure 3—figure supplement 1*), peak hCFR was 0.35 in the week beginning 8 March. There was a decline over the spring to a low of 0.17 in the week beginning 12 July, but this trend subsequently reversed and

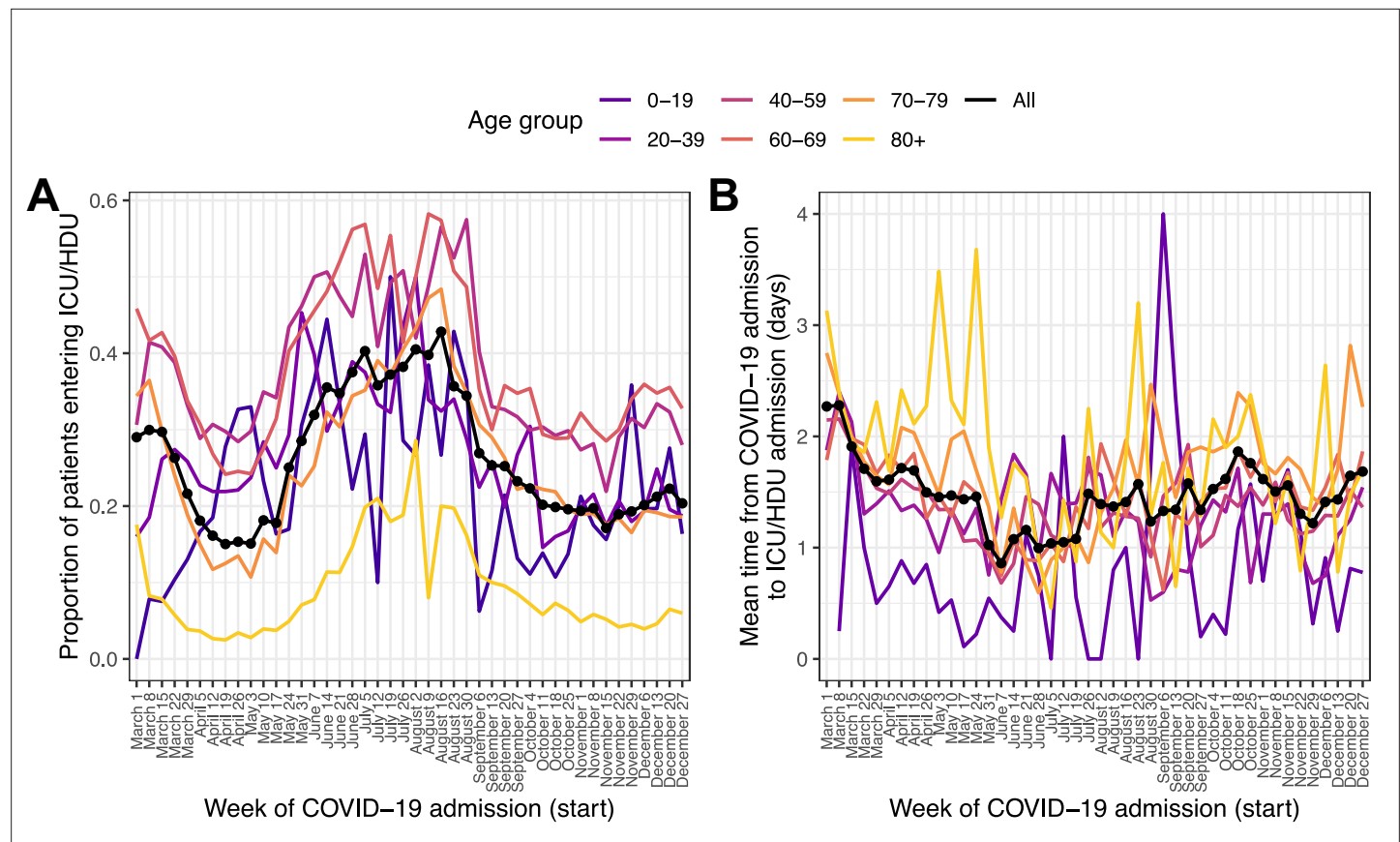

**Figure 2.** Patients entering ICU/HDU within 13 days of COVID-19 admission (**A**) and time from COVID-19 admission to ICU/HDU admission (**B**) over time. Each line is the proportion (**A**) or mean value (**B**) amongst all patients (black, dotted) or patients in each age group (coloured).

The online version of this article includes the following source data for figure 2:

**Source data 1.** Proportion of individuals entering ICU by week of COVID-19 admission, according to age group and overall.

**Source data 2.** Mean time in days from COVID-19 admission to ICU admission by week of COVID-19 admission, according to age group and overall.

reached 0.32 by mid-December. At the same time, the mean time from admission to outcome in this whole population showed very little change following a dramatic decline during March, from 16 days in the week beginning 1 March to 10 at the start of April (*Figure 3—figure supplement 2*). These overall patterns, however, mask substantial variation by on ICU/HDU admission and, in the latter case, outcome (*Figure 3*). The trend in hCFR is largely driven by patients who were not admitted to an ICU or HDU. The most consistent decline in time to outcome was observed in ICU/HDU admissions who survived (a decline in the mean of 7.6 days between the first and last weeks studied, *Figure 3B*, bottom left) while survivors with no ICU/HDU admission showed, as with the overall trend, little change after March (bottom right). Variation in time to death appeared very modest amongst patients with an ICU/HDU admission (top left), while there was a distinct peak around August and September in those without (top right). When age is also considered (*Figure 3—figure supplements 3 and 4*), a notable additional pattern is the clear correlation of time to discharge and age in surviving non-ICU/HDU patients, which is much less obvious, if present at all, in patients with an ICU/HDU admission.

The results of the three multivariable regression analyses can be seen in *Table 4*; some variables (country and the 'unknown' class for comorbidities) are excluded for brevity, but the full version is provided as *Supplementary file 7*. Note that all variables are adjusted for all others, which is also the case for all the other regressions presented in this paper. There was strong evidence of an association of month of COVID-19 admission with all three variables (Wald test p < 0.001 in all cases).

Amongst non-ICU/HDU patients, the month with the greatest odds of death was March (OR 1.12 compared to April, 95% CI 1.07–1.17), while that with the smallest was July (OR 0.35, 95% CI 0.29–0.43). April (the reference category) was the month with the shortest time to death, while August had the longest (54.2 % increase, 95% CI 32.9–78.8%). Variation in time to discharge was more modest; the month with the largest value of this variable was March (3.7 % increase, 95% CI 1.9–5.5%), and that with the shortest was August (12.3 % decrease, 95% CI 7.2–17.1%).

ICU/HDU admission was associated with a 7.56-fold higher odds of death (95% CI 6.81–8.4), a 81.8 % increase in time to death (95% CI 70.3–94.2%) and a 171.6 % increase in time to discharge (95% CI 162.3–181.3%). As a result of the patterns observed in *Figure 3*, we also fitted interaction terms of month of COVID-19 admission with IDU/HCU admission. Their inclusion was consistently statistically significant (Wald test p < 0.001 in all cases), although for odds of death this ceases to be true when December is removed (p = 0.23). Hence, the overall increased odds of death amongst ICU/HDU patients was significantly mitigated in December (combined OR 6.07 vs non-ICU/HDU admissions in December, 95% CI 5.23–7.06). There was no evidence that ICU/HDU patients admitted in March, May, or August had a longer time to death than April, but the estimates for all other months were significantly greater, with the peak in November (21.5 % increase, 95% CI 15.1–28.2%). The longest times to discharge in these patients were in March (4.2 % increase vs April, 95% CI 0.8–7.8%) and the shortest in December (28.1 % decrease, 95% CI 24–32%).

Increasing age was associated with monotonic increases in odds of death and time to discharge, with and without ICU/HDU admission. Time to death showed little evidence of variation by age in non-ICU/HDU patients except for marginal evidence for an increase in the oldest age group (5.7 % increase vs 40–59, 95% CI 0.3–11.2%). In ICU/HDU patients, however, where an interaction term was again fitted, the shortest times to death were recorded in both the youngest (49.2% decrease, 95% CI 35.8–59.9%) and oldest (27.6 % decrease, 95% CI 24.2–30.9%) groups; longest times to death were in middle-aged adults (40-69). Male sex was associated with higher odds of death (OR 1.33, 95% CI 1.29–1.38), and small increases in time to both death (3.4 % increase, 95% CI 1.7–5.2%) and discharge (1.6 % increase, 95% CI 0.5–2.7%). Symptom onset following admission was also associated with higher odds of death (OR 1.28, 95% CI 1.21–1.35) and large increases in time to death (24.4 % increase, 21.1–27.8%) and discharge (53.2 % increase, 95% CI 49.7–56.7%). Patients admitted more than a week from symptom onset had lower odds of death, and shorter stays in hospital, regardless of outcome (see *Table 4*). Where associations with comorbidities were detected, the majority were in the direction of poorer outcomes (increased hCFR, decreased time to death, and increased time to discharge), with a few exceptions. Most notably, asthma was associated with lower odds of death (OR 0.93, 95% CI 0.88–0.97), longer times to death (2.9 % increase, CI 0.2–5.6%) and shorter times to discharge (1.9 % decrease, 95% CI 0.3–3.4%).

To further illustrate these findings, *Figure 4* displays time trends in model predictions for hCFR, time to death and time to discharge for typical patients of both sexes in every age group, both for

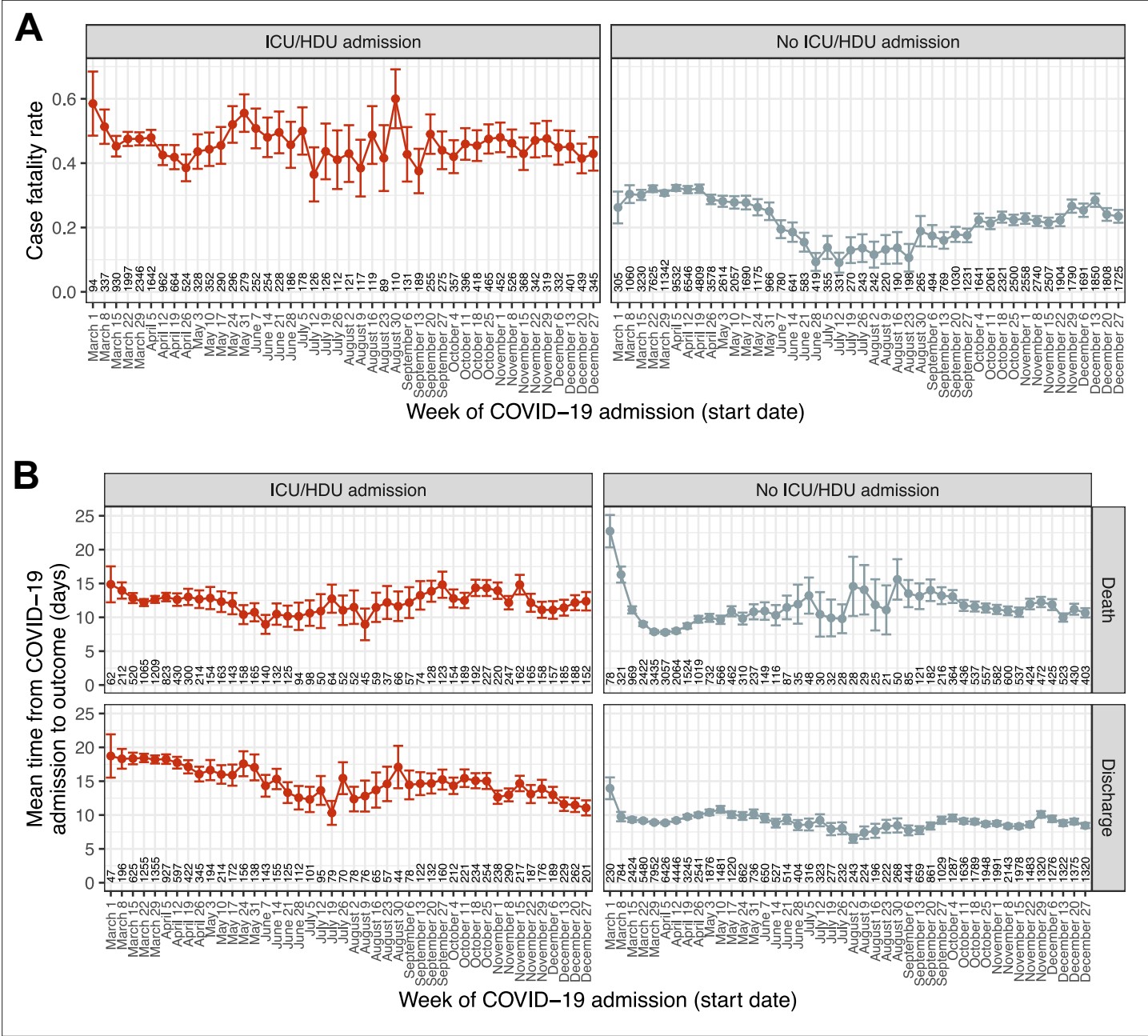

**Figure 3.** Temporal trends in outcome and time to outcome. (**A**) Case fatality ratio in patients experiencing death or discharge within 45 days of COVID-19 admission, by recorded ICU/HDU admission. (**B**) Mean time from COVID-19 admission to the outcome of death or discharge, further faceted by ICU/HDU admission. Error bars represent 95 % confidence intervals. Numbers along the x-axis indicate the numbers of patients involved in each category.

The online version of this article includes the following source data and figure supplement(s) for figure 3:

**Source data 1.** Estimate and 95 % confidence interval for hCFR by week of COVID-19 admission, according to ICU/HDU admission status.

**Source data 2.** Estimate and 95 % confidence interval for time from COVID-19 admission to outcome by week of COVID-19 admission, according to ICU/HDU admission status and outcome.

**Source data 3.** Estimated overall hCFR by week of COVID-19 admission.

**Source data 4.** Estimated mean time from COVID-19 admission to outcome by week of COVID-19 admission.

**Source data 5.** Estimate and 95 % confidence interval for hCFR by week of COVID-19 admission, according to ICU/HDU admission status and age group.

**Source data 6.** Estimate and 95 % confidence interval for time from COVID-19 admission to outcome by week of COVID-19 admission, according to

*Figure 3 continued on next page*

*Figure 3 continued*

ICU/HDU admission status and outcome, and age group.

**Figure supplement 1.** Temporal trends in case fatality rate amongst all patients.

**Figure supplement 2.** Temporal trends in mean time from COVID-19 admission to final outcome (death or discharge).

**Figure supplement 3.** Temporal trends in case fatality rate, faceted by ICU/HDU admission and further separated by age group.

**Figure supplement 4.** Temporal trends in mean time from COVID-19 admission to final outcome, faceted by outcome and ICU/HDU admission and further separated by age group.

those with disease serious enough to trigger ICU/HDU admission and those without. These patients were assumed to be admitted in the UK after less than a week of symptoms. Each was assigned the median number of comorbidities for their combination of sex and age group in the real dataset, and the exact comorbidities chosen were also the most common in that demographic group; see inset table, *Figure 4*. (For example, males in the 60–69 age group had a median of two comorbidities recorded, and the two most common were hypertension and chronic cardiac disease.)

## Status by days since admission

*Figure 5* displays Sankey diagrams reflecting the location of patients within hospital (ward or ICU) or their final status (dead, discharged, or unknown) on the day of COVID-19 admission (**A**), 3 days later (A + 3), 7 days later (A + 7) or, to represent the final status only, 1 day after the last day in hospital (O + 1). The plot is facetted by age group and month of COVID-19 admission. For simplicity, only four months (April, June, August and October) appear in the main figure, but see *Figure 5—figure supplement 1* for all months, featuring a total of 129,044 patients (90.5%).

## Discussion

To the best of our knowledge and at the time of publication, this is the largest international cohort of COVID-19 patients in the world. Considerable temporal variations in the events preceding and during hospitalisation for patients with confirmed COVID-19 were observed during the period March to December 2020. We specifically looked at length of illness before admission, probability of ICU/HDU admission, time to ICU/HDU admission for those so admitted, case fatality rate, and duration of admission overall.

These results highlight key findings with practical implications for case management, resource allocation, performance benchmarking, and reporting of outcomes in research, and point to the fact that patients' journeys vary over time and must be interpreted with the background of transmission intensity, policy, and practice where cases occur. Therefore, static 'snapshots' of the situation at any one time may lead to misguided practice and management if not regularly monitored and approaches adapted accordingly.

In a recent preprint (*Kirwan et al., 2021*), analysed temporal variation on time from hospital admission to death, discharge or ICU/HDU admission amongst a smaller cohort of UK patients. We confirm many of the trends that they identified, including the lower hCFRs over the summer and the increased odds of ICU/HDU admission in middle-aged age groups. They did not, however, detect the increase in the proportion of patients with an ICU/HDU admission during the summer, or the decline in time to discharge amongst non-ICU/HDU patients over the entire time period. As there were many fewer hospitals included in that study than in ours (31 vs 620) this may be suggestive of variation in available ICU/HDU capacity and usage amongst participating sites in the two studies.

### Prior to admission

Across all age groups, the length of illness before seeking hospital care was longest in July and August when case numbers were lower, and shortest at the extremities of the age distribution and for females. The latter variations may, at least partially, be explained by differences in health-care-seeking behaviour by different demographics, and by differences in clinical progression of disease for different groups. Along similar lines, the fact that patients who died had consistently shorter duration of illness before hospital admission may reflects the fact that more serious cases evolve more rapidly and those affected seek care earlier. In this scenario, patients admitted after experiencing symptoms for longer

**Table 4.** Combined results of a logistic regression analysis identifying predictors of death as an outcome, and two linear regression analyses identifying correlates of time to death and time to discharge.

All analyses are multivariable. For brevity, the country variable, as well as the 'unknown' class for each comorbidity (representing patients with missing data for that condition) are omitted here; see *Supplementary file 7* for a version with them included. The p-values of Wald tests for the inclusion of each variable in each regression are included as a separate column; these were calculated including the 'unknown' class for comorbidities.

| | Odds ratio (death v discharge) | | | Time to death (% change, days) | | | Time to discharge (% change, days) | | |
|---|---|---|---|---|---|---|---|---|---|
| | Estimate | 95 % confidence interval | Wald test p-value | Estimate | 95 % confidence interval | Wald test p-value | Estimate | 95 % confidence interval | Wald test p-value |
| Month of COVID admission (ref: April) | | | < 0.001 | | | < 0.001 | | | < 0.001 |
| March | 1.1 | (1.1, 1.2) | | 14.7 | (12.2, 17.3) | | 3.7 | (2.1, 5.2) | |
| May | 0.7 | (0.7, 0.8) | | 15.5 | (12.1, 19.1) | | 1.3 | (–0.6, 3.3) | |
| June | 0.5 | (0.5, 0.6) | | 20.8 | (14.2, 27.7) | | –2.8 | (–5.6, –0.02) | |
| July | 0.3 | (0.3, 0.4) | | 28.1 | (14.9, 42.8) | | –8.5 | (–12.2, –4.7) | |
| August | 0.4 | (0.3, 0.5) | | 47.2 | (29.5, 67.3) | | –10.8 | (–15.0, –6.4) | |
| September | 0.6 | (0.5, 0.6) | | 40.7 | (32.6, 49.3) | | –3.2 | (–5.8, –0.5) | |
| October | 0.6 | (0.6, 0.7) | | 33.2 | (28.9, 37.7) | | –1.4 | (–3.2, 0.4) | |
| November | 0.6 | (0.6, 0.7) | | 26.7 | (22.8, 30.8) | | –3.5 | (–5.2, –1.8) | |
| December | 0.8 | (0.8, 0.9) | | 26.3 | (22.3, 30.4) | | 2.2 | (0.2, 4.2) | |
| Age group (ref: 40–59) | | | < 0.001 | | | < 0.001 | | | < 0.001 |
| 10–19 | 0.3 | (0.2, 0.4) | | –0.5 | (–24.3, 30.6) | | –33.6 | (–35.7, –31.4) | |
| 20–39 | 0.3 | (0.2, 0.3) | | –1.7 | (–15.9, 14.8) | | –16.9 | (–18.5, –15.3) | |
| 60–69 | 2.9 | (2.7, 3.2) | | –2.1 | (–6.8, 3.0) | | 17.4 | (15.6, 19.2) | |
| 70–79 | 6.1 | (5.7, 6.6) | | 1.9 | (–2.6, 6.6) | | 36.4 | (34.3, 38.6) | |
| 80+ | 10.9 | (10.1, 11.8) | | 4.3 | (–0.3, 9.0) | | 52.9 | (50.4, 55.4) | |

*Table 4 continued on next page*

*Table 4 continued*

| | Odds ratio (death v discharge) | | | Time to death (% change, days) | | | Time to discharge (% change, days) | | |
|---|---|---|---|---|---|---|---|---|---|
| | Estimate | 95 % confidence interval | Wald test p-value | Estimate | 95 % confidence interval | Wald test p-value | Estimate | 95 % confidence interval | Wald test p-value |
| ICU/HDU admission | 7.6 | (6.8, 8.4) | < 0.001 | 68.3 | (59.1, 78.0) | < 0.001 | 140.1 | (133.0, 147.3) | < 0.001 |
| Sex (ref: female) | | | < 0.001 | | | < 0.001 | | | 0.0059 |
| Male | 1.3 | (1.3, 1.4) | | 2.9 | (1.4, 4.5) | | 1.2 | (0.2, 2.1) | |
| Pregnant (ref: female, no) | | | 0.097 | | | 0.069 | | | < 0.001 |
| Female, yes | 0.6 | (0.4, 1.0) | | –5.4 | (–27.4, 23.3) | | –18.6 | (–22.3, –14.6) | |
| Days from symptom onset to hospital admission (ref: 0–6) | | | < 0.001 | | | < 0.001 | | | < 0.001 |
| Symptom onset post-admission | 1.3 | (1.2, 1.3) | | 20.7 | (17.9, 23.5) | | 45.7 | (42.9, 48.6) | |
| 7–13 | 0.7 | (0.7, 0.8) | | –1.7 | (–3.5, 0.1) | | –3.6 | (–4.6, –2.5) | |
| 14+ | 0.7 | (0.7, 0.8) | | –0.2 | (–2.9, 2.5) | | –6.1 | (–7.6, –4.5) | |
| Comorbidities | | | | | | | | | |
| Asthma | 0.9 | (0.9, 1.0) | < 0.001 | 2.5 | (0.2, 4.9) | 0.11 | –1.6 | (–2.9, –0.3) | 0.048 |
| Chronic cardiac disease | 1.2 | (1.2, 1.3) | < 0.001 | –2.8 | (–4.4, –1.3) | < 0.001 | 1.3 | (0.1, 2.6) | 0.075 |
| Chronic haemotologic disease | 1.2 | (1.1, 1.3) | < 0.001 | 0.2 | (–3.1, 3.6) | 0.62 | 5.8 | (3.1, 8.6) | < 0.001 |
| Chronic kidney disease | 1.4 | (1.4, 1.5) | < 0.001 | –2.4 | (–4.1, –0.6) | 0.012 | 5.8 | (4.2, 7.4) | < 0.001 |
| Chronic neurological disorder | 1.4 | (1.3, 1.4) | < 0.001 | –0.7 | (–2.7, 1.4) | 0.61 | 15.3 | (13.4, 17.2) | < 0.001 |

*Table 4 continued on next page*

*Table 4 continued*

| | Odds ratio (death v discharge) | | | Time to death (% change, days) | | | Time to discharge (% change, days) | | |
|---|---|---|---|---|---|---|---|---|---|
| | Estimate | 95 % confidence interval | Wald test p-value | Estimate | 95 % confidence interval | Wald test p-value | Estimate | 95 % confidence interval | Wald test p-value |
| Chronic pulmonary disease | 1.4 | (1.3, 1.4) | < 0.001 | −2.2 | (−3.9, −0.5) | 0.034 | 3.9 | (2.4, 5.3) | < 0.001 |
| Dementia | 1.5 | (1.4, 1.6) | < 0.001 | −1.2 | (−3.1, 0.8) | 0.43 | 9 | (7.0, 11.0) | < 0.001 |
| Diabetes | 1.1 | (1.1, 1.2) | < 0.001 | −4.2 | (−6.0, −2.3) | < 0.001 | 1.8 | (0.4, 3.2) | < 0.001 |
| HIV/AIDS | 1.2 | (1.0, 1.6) | 0.005 | 4 | (−8.2, 17.8) | 0.58 | 2.2 | (−5.1, 10.1) | 0.0025 |
| Hypertension | 1 | (0.9, 1.0) | < 0.001 | 0.2 | (−1.5, 1.9) | 0.14 | 2 | (0.8, 3.2) | < 0.001 |
| Liver disease | 1.4 | (1.3, 1.6) | < 0.001 | 5.2 | (1.1, 9.5) | < 0.001 | 12 | (9.0, 15.1) | < 0.001 |
| Malignant neoplasm | 1.4 | (1.4, 1.5) | < 0.001 | 3.5 | (1.2, 5.8) | 0.0059 | 2.2 | (0.3, 4.1) | < 0.001 |
| Malnutrition | 1.3 | (1.2, 1.5) | < 0.001 | 0.6 | (−3.6, 5.0) | 0.81 | 11.5 | (7.5, 15.7) | < 0.001 |
| Obesity | 1.1 | (1.1, 1.2) | < 0.001 | −5.5 | (−7.7, −3.2) | < 0.001 | 6.3 | (4.8, 7.9) | < 0.001 |
| Rheumatological disorder | 1 | (0.9, 1.0) | 0.045 | 2.3 | (0.1, 4.7) | 0.052 | 0.6 | (−1.0, 2.2) | 0.44 |
| Smoking | 1.1 | (1.1, 1.2) | < 0.001 | 4.2 | (0.5, 8.0) | 0.1 | 2.2 | (0.04, 4.4) | 0.034 |
| Interaction: ICU/HDU admission _ month of admission (ref: April) | | | < 0.001 | | | < 0.001 | | | < 0.001 |
| March | 1 | (0.9, 1.1) | | −14.3 | (−18.1, −10.4) | | 0.6 | (−3.1, 4.5) | |
| May | 1.1 | (0.9, 1.3) | | −12 | (−17.6, −6.1) | | −9 | (−13.9, −3.9) | |
| June | 1.2 | (1.0, 1.5) | | −9.1 | (−16.8, −0.8) | | −11.3 | (−17.1, −5.1) | |
| July | 1.4 | (1.1, 1.9) | | −14.8 | (−25.8, −2.1) | | −18.8 | (−25.2, −11.8) | |
| August | 1.1 | (0.8, 1.6) | | −27.9 | (−38.4, −15.6) | | −15.6 | (−23.2, −7.3) | |
| September | 1.1 | (0.9, 1.4) | | −16.6 | (−24.1, −8.4) | | −6.9 | (−13.0, −0.3) | |
| October | 1.1 | (1.0, 1.3) | | −13.7 | (−18.9, −8.2) | | −11.6 | (−15.9, −7.0) | |
| November | 1.1 | (1.0, 1.3) | | −6.6 | (−12.1, −0.7) | | −15.9 | (−20.0, −11.6) | |

*Table 4 continued on next page*

*Table 4 continued*

| | Odds ratio (death v discharge) | | | Time to death (% change, days) | | | Time to discharge (% change, days) | | |
|---|---|---|---|---|---|---|---|---|---|
| | Estimate | 95 % confidence interval | Wald test p-value | Estimate | 95 % confidence interval | Wald test p-value | Estimate | 95 % confidence interval | Wald test p-value |
| December | 0.8 | (0.7, 0.9) | | −11.6 | (−17.0, −5.8) | | −27.9 | (−31.4, −24.2) | |
| Interaction: ICU/HDU admission _ age group (ref: 40–59) | | | < 0.001 | | | < 0.001 | | | < 0.00 |
| 10–19 | 0.8 | (0.4, 1.4) | | −42.8 | (−59.3, −19.6) | | 1.7 | (−6.0, 10.2) | |
| 20–39 | 1.8 | (1.4, 2.4) | | −11.8 | (−25.9, 4.9) | | 0.8 | (−3.6, 5.4) | |
| 60–69 | 0.7 | (0.7, 0.8) | | 2.5 | (−3.7, 9.0) | | −7.2 | (−10.4, −3.9) | |
| 70–79 | 0.6 | (0.5, 0.7) | | −11.8 | (−16.8, −6.6) | | −21.8 | (−24.9, −18.5) | |
| 80+ | 0.4 | (0.3, 0.5) | | −28.4 | (−32.9, −23.6) | | −40.1 | (−43.6, −36.4) | |
| Observations | 102,147 | | | 31,250 | | | 70,897 | | |

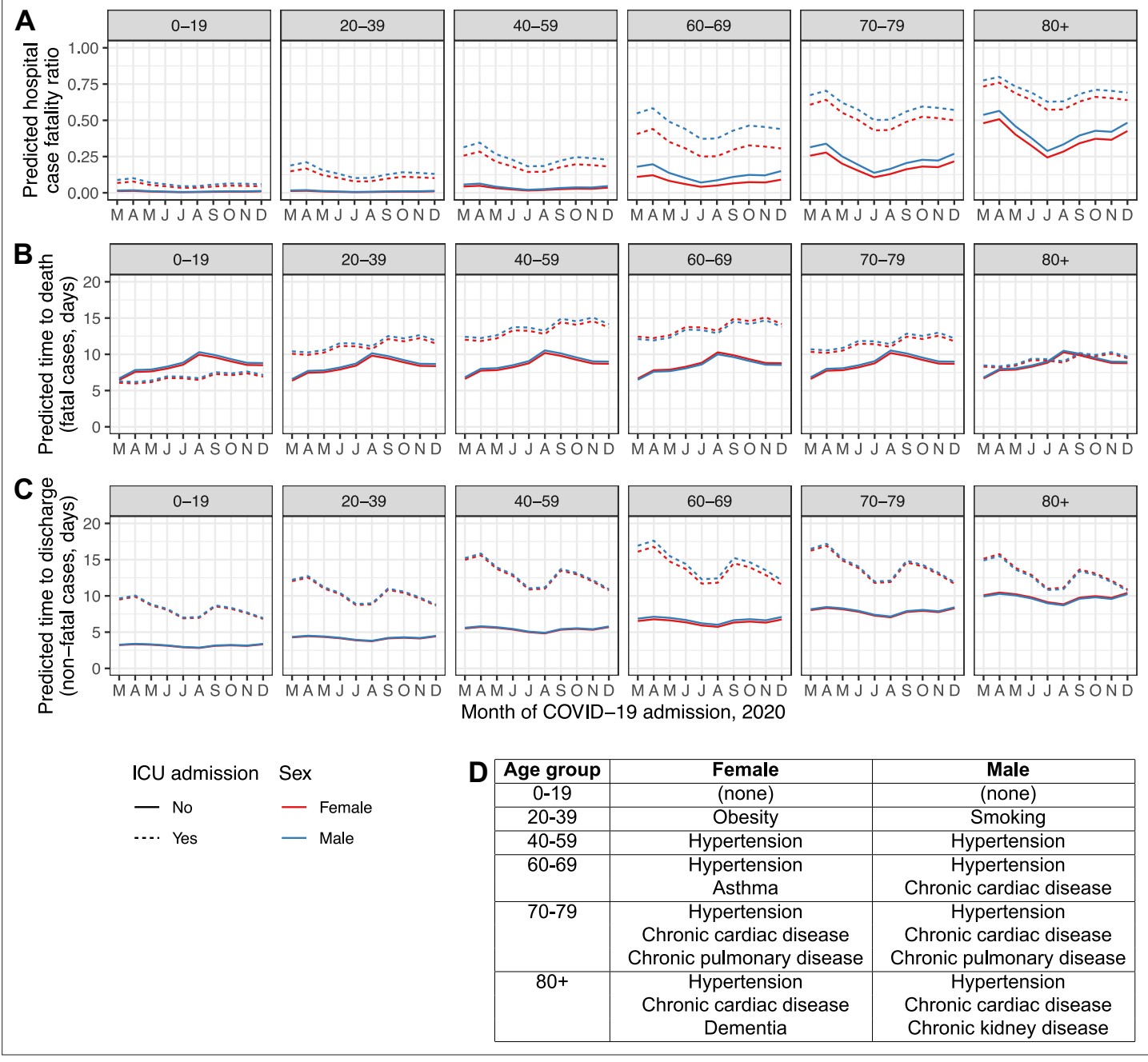

**Figure 4.** Regression model predictions for hospital CFR (**A**), predicted time to death in fatal cases (**B**) and predicted time to discharge in non-fatal cases (**C**) in a set of hypothetical typical patients. Lines are plotted by month of COVID-19 admission (y-axis), age group (facets, left to right), sex (red: female, blue: male), and ICU admission (solid lines: at least once, dotted lines: never). The inset table (**D**) lists the comorbidities assigned to the individuals in each combination of sex and age group.

The online version of this article includes the following source data for figure 4:

**Source data 1.** Predicted hCFR, time to death and time to discharge for all hypothetical patients.

than 1 week would be less likely to die because they were less serious cases and the individuals acted accordingly. The peak in time to admission during late summer and early autumn in the Northern Hemisphere may reflect delayed presentation following return from holiday, particularly given the high proportion of UK patients in this dataset and known viral importations to the UK from continental Europe around that time (*Hodcroft et al., 2021*).

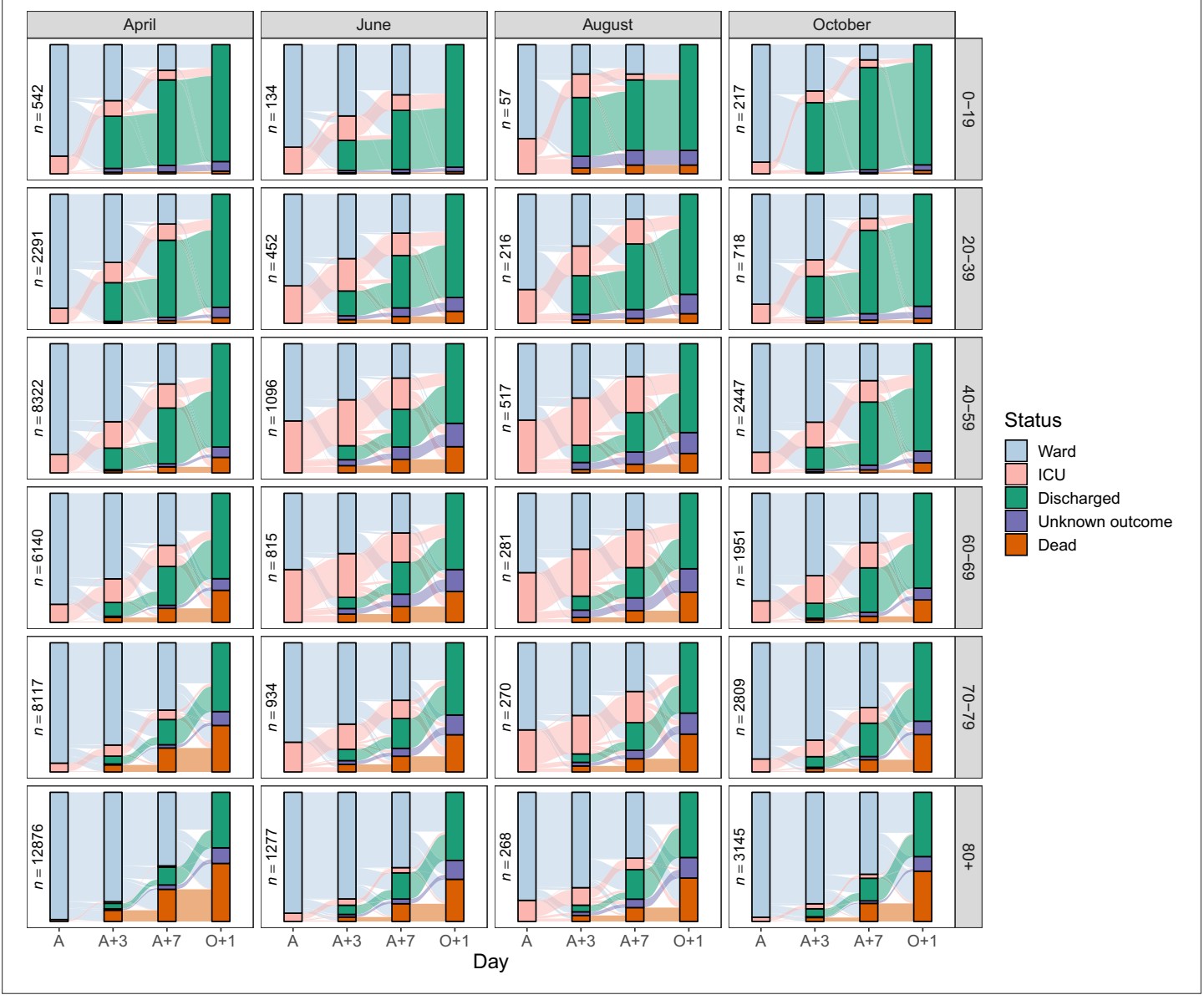

**Figure 5.** Sankey diagrams depicting the progress through the inpatient journey for patients with COVID-19 admission in April, June, August and October 2020, and subdivided by age. Bars are presented for the day of admission (A), 3 and 7 days later (A + 3 and A + 7), and the day after final outcome (O + 1).

The online version of this article includes the following source data and figure supplement(s) for figure 5:

**Source data 1.** Number of patients occupying a ward bed, occupying an ICU/HDU bed, dead, discharged and with unknown outcome on the day of admission (A), 3 and 7 days later (A + 3 and A + 7), and the day after final outcome (O + 1), by age group and month of COVID-19 admission.

**Figure supplement 1.** Expanded version of *Figure 5*, showing Sankey diagrams for all months.

At the same time, when considering the four most frequent symptoms at admission (fever, short of breath, cough, or fatigue), more symptoms were associated with a longer period between onset of symptoms and admission – and this was consistently so across the entire period under observation. This could be ascribed at least in part to variations in individual behaviour; some patients may present to hospital with a single symptom while others may wait a longer period until several have emerged. In addition, these phenomena could also partially be attributed to how case definitions are applied by physicians, or to the patient's own perceptions, or to those of their families. Some presentations are likely to be more alarming to the latter two groups than others; for example, individuals with none of

the four symptoms described above were admitted fastest of all and, amongst these, confusion was the most prevalent other symptom.

## During hospitalisation

Treating variables such as final outcome, ICU/HDU admission, or length of stay, as variables that remain static throughout an evolving epidemic is problematic, as demonstrated by our analyses. To give three examples: first, the case fatality rate showed an overall decline from 0.35 for cases admitted in March to 0.21 in July, followed by a renewed increase to 0.29 in December (*Figure 3—figure supplement 1*). Second, the data underlying the alluvial plots (*Figure 5*) allow us to determine that the proportion of patients discharged within a week of admission rose from 0.24 in March to a peak of 0.34 in September. Third, the proportion of still-admitted patients occupying an ICU/HDU bed showed considerable variation: for example, at day three this went from 0.19 in March to 0.13 in April, then rose to a peak of 0.38 in August before declining again, reaching a low of 0.15 in November. Variations in clinical care, the influence of treatments, and changes in available bed capacity are all likely to account for many of these differences. In older patients, the availability of social care space is another important variable.

Patients older than 80 had odds of being admitted to ICU/HDU over eight times smaller than those in the 40–59 category, which may reflect prognosis and the expected benefits of ICU/HDU admission, as well as patient preferences. Many serious chronic conditions were also associated with decreased odds, independently of age, likely for similar reasons. These decreased odds are also reflective of the temporality of the data. March and April represent our data's highest volume, which might reflect hospital capacity and the necessity for ICU/HDU prioritisation. For the patients who were admitted to ICU/HDU, there was no clear trend in the time from hospital admission to transfer to ICU/HDU after March. Length of illness before admission to the hospital and young age were associated with a shorter time from hospital admission to ICU/HDU (for example, a 9.2 % decrease for those waiting 7–13 days from onset compared to those waiting less than a week, and 32 % decrease amongst under-20s compared to the 40–59 age group), while a smaller proportion of older patients are escalated to ICU/HDU (OR 0.51 for ICU/HDU admission in the 70–79 age group) and after a longer time spent in the ward (a 4.2 % increase in the same age group).

## Outcome

As mentioned above, in patients with an outcome of death or discharge, hCFRs decreased from 0.35 in March to 0.21 by mid-2020 to increase again to 0.29 in December, mostly following the waves in the pandemic and therefore the number of admissions. System capacity may be an important predictor of patient outcome and may supersede other factors such as increasing case management skills and the influence of new therapies. This also warns against using outcome data that are not adequately controlled to assess efficacy and safety of treatments or other interventions, as effects may rather reflect capacity of a system to provide high-quality care.

We found that shorter time to death is associated with female sex, lack of ICU/HDU admission, and, amongst ICU/HDU patients, the extremes of age. Shorter time to discharge is also associated with female sex and lack of ICU/HDU admission, and this variable increases monotonically with age.

The finding of an association of asthma with reduced disease severity in COVID-19 is not unique to this study (*Alberca et al., 2021*; *Matsumoto and Saito, 2020*), but is also not a universal finding (*Choi et al., 2021a*; *Choi et al., 2021b*). A number of possible mechanisms for a protective effect have been suggested, including reduced ACE2 expression in the airway (*Jackson et al., 2020*), eosinophilia (*Ferastraoaru et al., 2021*), or simply the existing use of beneficial corticosteroids in this population (*Halpin et al., 2020*).

Cautionary notes in interpreting these findings. First, the dataset analysed is made of patients on the more severe end of the spectrum of disease compared to cases occurring in the community. Second, about half of these patients were hospitalised in just 2 months (March-April), were predominantly from the UK, and about half were over 70 years old. These demographics explain the high raw hCFR and the large proportion of patients presenting with age-related comorbidities – nearly half have hypertension, one-in-five chronic cardiac disease, and one-in-six diabetes. The regression results here should, however, be quite generalisable to hospitalised populations worldwide as country was accounted for as a predictor. Third, there are inherent limitations of observational data, however

large the dataset; in particular, we cannot attribute a cause to many of the phenomena described here. It is most notably not entirely possible to unpick biological effects from clinical decisions. As one example, the association of ICU/HDU admission with male sex is may be due not just to increased disease severity amongst males, but also clinician knowledge of the potential for more severe disease. Similarly, we see a lower rate of ICU/HDU admissions amongst individuals whose symptoms started following admission. On the one hand, it is likely that the population of patients with symptoms emerging in hospital had on average less severe disease, as mild community-acquired infections are less likely to present to hospital. On the other, as those patients will receive clinical care starting at the moment of diagnosis, the need for ICU/HDU is likely reduced even in more serious cases. Fourth, some variables are based on patient self-report which can be inexact; for example, it can be clearly seen in *Figure 1a* that multiples of seven reported days from symptom onset to admission are over-represented, suggesting reports in units of weeks. Fifth, some variables are not available to us; for example, resuscitation status and suitability for intensive care admission was not collected in our cohort, and without those variables the reasons for death or lack of ICU/HDU admission cannot be entirely unpicked. Similarly, we are not aware of what resource or bed capacity constraints may have affected individual sites at different times. Sixth, there may be a selection bias with respect to calendar time as a result of case volumes. Recruitment was performed by sites, upon identification of a patient with COVID-19 symptoms, according to their capacity, which was determined by the availability of staff to invite informed consent (as applicable) and complete data forms. Capacity will be subject to both geographical and temporal variations, and it is likely that both the proportion of patients recruited and the proportion completing follow-up would be reduced at times of high pressure on the site and the national healthcare system. However, enrolment was prospective, and as such staff would be blind to our outcomes of interest. In addition, while it is possible that individual sites chose patients to recruit (or cease following up) based on clinical characteristics, it is unclear why the basis for these decisions would show a consistent direction of bias amongst diverse locations. Lastly, one should refrain from overinterpreting data: some of the changes observed reflect adjustments in practice and logistics, or combined pressure on health systems, more than actual effects of interventions.

## Implications of findings

Often, in high-income countries, patient outcomes are seen through the lens of individualised treatment provided at the clinician patient interface. This paper demonstrates that outbreak epidemiology has an important influence on patient outcomes – the patient journey from likelihood of admission, through to disposition and length of stay in hospital, and overall outcome, change over the course of a pandemic. There are various explanations for variability – systems may at times be overwhelmed and unable to provide the usual quality of care to their patients; patient behaviour may change depending on perceptions of the status of the outbreak and the performance of the healthcare system at a given time; clinician familiarity with management of patients may vary; and changes in transmissibility and virulence are expected to occur.

The observed variability should inform on the limitations of using observational data during a long-lasting pandemic for management purposes in practice, and also question the use of some variables, such as length of stay in hospital or in ICU, as clinical trial outcomes. This demonstrates the importance of controlling for patient outcome data when designing clinical trials; for example, using our data, assessing a new treatment during the months of March to July will have shown a decrease in hCFR from 33%, to 21 % that may have been falsely attributed to a treatment effect without a concurrent randomised control.

At the same time, these findings also highlight the need for preparedness and resilience; the crucial importance of pre-positioned observational data collection systems that are adhered upon by a representative number of sites and are maintained for as long as the pandemic lasts; and the need for such capacity to be kept in-between epidemics.

## Acknowledgements

This work was supported by the UK Foreign, Commonwealth and Development Office and Wellcome [215091/Z/18/Z] and the Bill & Melinda Gates Foundation [OPP1209135]; CIHR Coronavirus Rapid Research Funding Opportunity OV2170359; Grants from Rapid European COVID-19 Emergency Response research (RECOVER) [H2020 project 101003589] and European Clinical Research

Alliance on Infectious Diseases (ECRAID) [965,313]; The Imperial NIHR Biomedical Research Centre; The Cambridge NIHR Biomedical Research Centre; and Endorsed by the Irish Critical Care- Clinical Trials Group, co-ordinated in Ireland by the Irish Critical Care- Clinical Trials Network at University College Dublin and funded by the Health Research Board of Ireland [CTN-2014–12]. This work uses Data / Material provided by patients and collected by the NHS as part of their care and support #DataSavesLives. The Data / Material used for this research were obtained from ISARIC4C. The COVID-19 Clinical Information Network (CO-CIN) data was collated by ISARIC4C Investigators. Data and Material provision was supported by grants from: the National Institute for Health Research (NIHR; award CO-CIN-01), the Medical Research Council (MRC; grant MC_PC_19059), and by the NIHR Health Protection Research Unit (HPRU) in Emerging and Zoonotic Infections at University of Liverpool in partnership with Public Health England (PHE), (award 200907), Wellcome Trust [Turtle, Lance-fellowship 205228/Z/16/Z], NIHR HPRU in Respiratory Infections at Imperial College London with PHE (award 200927), Liverpool Experimental Cancer Medicine Centre (grant C18616/A25153), NIHR Biomedical Research Centre at Imperial College London (award IS-BRC-1215–20013), and NIHR Clinical Research Network providing infrastructure support. This work was possible due to the dedication and hard work of the Norwegian SARS-CoV-2 study team, and supported by grants from Research Council of Norway grant no 312780, and a philanthropic donation from Vivaldi Invest A/S owned by Jon Stephenson von Tetzchner; The dedication and hard work of the Groote Schuur Hospital Covid ICU Team, and supported by the Groote Schuur nursing and University of Cape Town registrar bodies coordinated by the Division of Critical Care at the University of Cape Town; and supported by the COVID clinical management team, AIIMS, Rishikesh, India. Matthew Hall and Christophe Fraser were supported by a Li Ka Shing Foundation award to Christophe Fraser.

## Additional information

### Group author details

ISARIC Clinical Characterisation Group

Sheryl Ann Abdukahil; Ryuzo Abe; Laurent Abel; Lara Absil; Subhash Acharya; Andrew Acker; Shingo Adachi; Elisabeth Adam; Diana Adrião; Saleh Al Ageel; Shakeel Ahmed; Kate Ainscough; Ali Ait Hssain; Younes Ait Tamlihat; Takako Akimoto; Ernita Akmal; Eman Al Qasim; Razi Alalqam; Tala Al-dabbous; Senthilkumar Alegesan; Cynthia Alegre; Beatrice Alex; Kévin Alexandre; Abdulrahman Al-Fares; Huda Alfoudri; Imran Ali; Kazali Enagnon Alidjnou; Jeffrey Aliudin; Qabas Alkhafajee; Clotilde Allavena; Nathalie Allou; João Alves; João Melo Alves; Rita Alves; Maria Amaral; Heidi Ammerlaan; Phoebe Ampaw; Roberto Andini; Claire Andrejak; Andrea Angheben; François Angoulvant; Séverine Ansart; Massimo Antonelli; Carlos Alexandre Antunesde Brito; Ardiyan Apriyana; Yaseen Arabi; Irene Aragao; Francisco Arancibia; Carolline Araujo; Antonio Arcadipane; Patrick Archambault; Lukas Arenz; Jean-Benoît Arlet; Christel Arnold-Day; Lovkesh Arora; Rakesh Arora; Elise Artaud-Macari; Diptesh Aryal; Motohiro Asaki; Angel Asensio; Sheharyar Ashraf; Jean Baptiste Assie; Anika Atique; AMUdara Lakshan Attanyake; Johann Auchabie; Hugues Aumaitre; Adrien Auvet; Laurène Azemar; Cecile Azoulay; Benjamin Bach; Delphine Bachelet; Claudine Badr; Nadia Baig; J Kenneth Baillie; Erica Bak; Agamemnon Bakakos; Andriy Bal; Valeria Balan; Firouzé Bani-Sadr; Renata Barbalho; Wendy S Barclay; Michaela Barnikel; Helena Barrasa; Audrey Barrelet; Cleide Barrigoto; Marie Bartoli; Cheryl Bartone; Joaquín Baruch; Romain Basmaci; Denise Battaglini; Jules Bauer; Diego Fernando Bautista Rincon; Abigail Beane; Alexandra Bedossa; Sylvie Behilill; Aleksandr Beljantsev; David Bellemare; Anna Beltrame; Marine Beluze; Nicolas Benech; Dehbia Benkerrou; Suzanne Bennett; Luís Bento; Jan-Erik Berdal; Delphine Bergeaud; Hazel Bergin; José Luis Bernal Sobrino; Giulia Bertoli; Lorenzo Bertolino; Simon Bessis; Adam Betz; Sybille Bevilcaqua; Karine Bezulier; Amar Bhatt; Krishna Bhavsar; Isabella Bianchi; Claudia Bianco; Moirangthem Bikram Singh; Felwa Bin Humaid; François Bissuel; Patrick Biston; Laurent Bitker; Pablo Blanco-Schweizer; Catherine Blier; Frank Bloos; Mathieu Blot; Filomena Boccia; Laetitia Bodenes; Alice Bogaarts; Debby Bogaert; Anne-Hélène Boivin; Pierre-Adrien Bolze; François Bompart; Diogo Borges; Raphaël Borie; Hans Martin Bosse; Elisabeth Botelho-Nevers; Lila Bouadma; Olivier Bouchaud; Sabelline Bouchez; Dounia Bouhmani; Damien Bouhour; Kévin Bouiller; Laurence Bouillet; Camile Bouisse; Anne-Sophie Boureau; John Bourke; Maude

Bouscambert; Aurore Bousquet; Jason Bouziotis; Bianca Boxma; Marielle Boyer-Besseyre; Maria Boylan; Matthew Brack; Axelle Braconnier; Cynthia Braga; Timo Brandenburger; Filipa Brás Monteiro; Luca Brazzi; Dorothy Breen; Patrick Breen; Patrick Breen; Kathy Brickell; Alex Browne; Shaunagh Browne; Nicolas Brozzi; Nina Buchtele; Christian Buesaquillo; Marielle Buisson; Erlina Burhan; Ingrid G Bustos; André Cabie; Susana Cabral; Eder Caceres; Cyril Cadoz; Kate Calligy; Jose Andres Calvache; João Cam; Valentine Campana; Paul Campbell; Cecilia Canepa; Mireia Cantero; Pauline Caraux-Paz; Sheila Cárcel; Chiara Simona Cardellino; Filipa Cardoso; Filipe Cardoso; Nelson Cardoso; Sofia Cardoso; Simone Carelli; Nicolas Carlier; Thierry Carmoi; Gayle Carney; Chloe Carpenter; Inês Carqueja; Marie-Christine Carret; François Martin Carrier; Ida Carroll; Gail Carson; Ed Carton; Maire-Laure Casanova; Mariana Cascão; Siobhan Casey; José Casimiro; Bailey Cassandra; Silvia Castañeda; Nidyanara Castanheira; Guylaine Castor-Alexandre; Henry Castrillón; Ivo Castro; Ana Catarino; François-Xavier Catherine; Paolo Cattaneo; Roberta Cavalin; Giulio Giovanni Cavalli; Alexandros Cavayas; Adrian Ceccato; Minerva Cervantes-Gonzalez; Anissa Chair; Catherine Chakveatze; Adrienne Chan; Meera Chand; Christelle Chantalat Auger; Jean-Marc Chapplain; Julie Chas; Jonathan Samuel Chávez Iñiguez; Anjellica Chen; Yih-Sharng Chen; Matthew Pellan Cheng; Antoine Cheret; Thibault Chiarabini; Julian Chica; Catherine Chirouze; Davide Chiumello; Hwa Jin Cho; Sung-Min Cho; Bernard Cholley; Marie-Charlotte Chopin; JosePedro Cidade; Barbara Wanjiru Citarella; Anna Ciullo; Jennifer Clarke; Sara Clohisey; Necsoi Coca; Cassidy Codan; Caitriona Cody; Alexandra Coelho; Megan Coles; Gwenhaël Colin; Michael Collins; Sebastiano Maria Colombo; Pamela Combs; Jennifer Connolly; Marie Connor; Anne Conrad; Sofía Contreras; Elaine Conway; Graham S Cooke; Mary Copland; Hugues Cordel; Amanda Corley; Sarah Cormican; Sabine Cornelis; Arianne Joy Corpuz; Grégory Corvaisier; Camille Couffignal; Sandrine Couffin-Cadiergues; Roxane Courtois; Stéphanie Cousse; Sabine Croonen; Gloria Crowl; Jonathan Crump; Claudina Cruz; Juan Luis Cruz Berm; Jaime Cruz Rojo; Marc Csete; Alberto Cucino; Ailbhe Cullen; Caroline Cullen; Matthew Cummings; Ger Curley; Elodie Curlier; Colleen Curran; Paula Custodio; Anada Silva Filipe; Charlene Da Silveira; Al-Awwab Dabaliz; Andrew Dagens; Darren Dahly; Heidi Dalton; Jo Dalton; Seamus Daly; Federico D'Amico; Nick Daneman; Corinne Daniel; Emmanuelle A Dankwa; Jorge Dantas; Frédérick D'Aragon; Markde Boer; Gilliande Loughry; Diegode Mendoza; Etienne De Montmollin; Rafael Freitas de Oliveira França; Ana Isabel de Pinho Oliveira; De Rosanna Rosa; Thushande Silva; Peterde Vries; Jillian Deacon; David Dean; Alexa Debard; Bianca De Benedictis; Marie-Pierre Debray; Nathalie De Castro; William Dechert; Lauren Deconninck; Romain Decours; Eve Defous; Isabelle Delacroix; Eric Delaveuve; Karen Delavigne; Nathalie M Delfos; Ionna Deligiannis; Andrea Dell'Amore; Christelle Delmas; Pierre Delobel; Elisa Demonchy; Emmanuelle Denis; Dominique Deplanque; Pieter Depuydt; Mehul Desai; Diane Descamps; Mathilde Desvallée; Santi Dewayanti; Alpha Diallo; Sylvain Diamantis; André Dias; Juan Jose Diaz; Priscila Diaz; Rodrigo Diaz; Kévin Didier; Jean-Luc Diehl; Wim Dieperink; Jérôme Dimet; Vincent Dinot; Fara Diop; Alphonsine Diouf; Yael Dishon; Félix Djossou; Annemarie B Docherty; Helen Doherty; Arjen M Dondorp; Christl A Donnelly; Maria Donnelly; Chloe Donohue; Sean Donohue; Yoann Donohue; Ciara Doran; Peter Doran; Céline Dorival; Eric D'Ortenzio; James Joshua Douglas; Renee Douma; Nathalie Dournon; Triona Downer; Joanne Downey; Mark Downing; Tom Drake; Aoife Driscoll; Claudio Duarte Fonseca; Vincent Dubee; François Dubos; Alexandre Ducancelle; Toni Duculan; Susanne Dudman; Abhijit Duggal; Paul Dunand; Jake Dunning; Mathilde Duplaix; Emanuele Durante-Mangoni; Bertrand Dussol Lucian Durham III; Juliette Duthoit; Xavier Duval; Anne Margarita Dyrhol-Riise; Marco Echeverria-Villalobos; Siobhan Egan; Carla Eira; Mohammed El Sanharawi; Subbarao Elapavaluru; Brigitte Elharrar; Jacobien Ellerbroek; Philippine Eloy; Tarek Elshazly; Isabelle Enderle; Ilka Engelmann; Vincent Enouf; Olivier Epaulard; Martina Escher; Mariano Esperatti; Hélène Esperou; Marina Esposito-Farese; João Estevão; Manuel Etienne; Nadia Ettalhaoui; Anna Greti Everding; Mirjam Evers; Isabelle Fabre; Marc Fabre; Amna Faheem; Arabella Fahy; Cameron J Fairfield; Pedro Faria; Nataly Farshait; Arie Zainul Fatoni; Karine Faure; Raphaël Favory; Mohamed Fayed; Niamh Feely; Laura Feeney; Jorge Fernandes; Marília Fernandes; Susana Fernandes; François-Xavier Ferrand; Eglantine Ferrand Devouge; Joana Ferrão; Mário Ferraz; Benigno Ferreira; Sílvia Ferreira; Ricard Ferrer-Roca; Nicolas Ferriere; Céline Ficko; Claudia Figueiredo-Mello; Juan Fiorda; Thomas Flament; Clara Flateau; Tom Fletcher; Letizia Lucia Florio; Brigid Flynn; Deirdre Flynn; Claire Foley; Jean Foley; Tatiana Fonseca; Patricia Fontela; Simon Forsyth; Denise Foster; Giuseppe

Foti; Erwan Fourn; Robert A Fowler; Dr Marianne Fraher; Diego Franch-Llasat; Christophe Fraser; John F Fraser; Marcela Vieira Freire; Ana Freitas Ribeiro; Caren Friedrich; Ricardo Fritz; Stéphanie Fry; Nora Fuentes; Masahiro Fukuda; Valérie Gaborieau; Rostane Gaci; Massimo Gagliardi; Jean-Charles Gagnard; Amandine Gagneux-Brunon; Sérgio Gaião; Linda Gail Skeie; Phil Gallagher; Elena Gallego Curto; Carrol Gamble; Arthur Garan; Rebekha Garcia; Noelia García Barrio; Esteban Garcia-Gallo; Denis Garot; Valérie Garrait; Nathalie Gault; Aisling Gavin; Anatoliy Gavrylov; Alexandre Gaymard; Johannes Gebauer; Eva Geraud; Louis Gerbaud Morlaes; Nuno Germano; Jade Ghosn; Marco Giani; Carlo Giaquinto; Jess Gibson; Tristan Gigante; Morgane Gilg; Elaine Gilroy; Guillermo Giordano; Michelle Girvan; Valérie Gissot; Judy Gitahi; Gezy Giwangkancana; Daniel Glikman; Eric Gnall; Geraldine Goco; François Goehringer; Siri Goepel; Jean-Christophe Goffard; Jonathan Golob; Rui Gomes; Joan Gómez-Junyent; Marie Gominet; Alicia Gonzalez; Patricia Gordon; Isabelle Gorenne; Laure Goubert; Cécile Goujard; Tiphaine Goulenok; Margarite Grable; Jeronimo Graf; Edward Wilson Grandin; Pascal Granier; Giacomo Grasselli; Lorenzo Grazioli; Christopher A Green; William Greenhalf; Segolène Greffe; Domenico Luca Grieco; Matthew Griffee; Fiona Griffiths; Ioana Grigoras; Albert Groenendijk; Anja Grosse Lordemann; Heidi Gruner; Yusing Gu; Fabio Guarracino; Jérémie Guedj; Martin Guego; Dewi Guellec; Anne-Marie Guerguerian; Daniela Guerreiro; Romain Guery; Anne Guillaumot; Laurent Guilleminault; Maisa Guimarães de Castro; Thomas Guimard; Daniel Haber; Hannah Habraken; Ali Hachemi; Nadir Hadri; Olena Haidash; Sheeba Hakak; Adam Hall; Matthew Hall; Sophie Halpin; Ansley Hamer; Rebecca Hamidfar; Terese Hammond; Rashan Haniffa; Hayley Hardwick; Ewen M Harrison; Janet Harrison; Samuel Bernard Ekow Harrison; Alan Hartman; Muhammad Hayat; Ailbhe Hayes; Leanne Hays; Jan Heerman; Lars Heggelund; Ross Hendry; Martina Hennessy; Aquiles Henriquez-Trujillo; Maxime Hentzien; Jaime Hernandez-Montfort; Daniel Herr; Andrew Hershey; Liv Hesstvedt; Astarini Hidayah; Dawn Higgins; Eibhilin Higgins; Grainne Higgins OKeeffe; Rita Hinchion; Samuel Hinton; Hiroaki Hiraiwa; Hikombo Hitoto; Antonia Ho; Alexandre Hoctin; Isabelle Hoffmann; Oscar Hoiting; Rebecca Holt; Jan Cato Holter; Peter Horby; Juan Pablo Horcajada; Koji Hoshino; Kota Hoshino; Ikram Houas; Catherine L Hough; Stuart Houltham; Jimmy Ming-Yang Hsu; Jean-Sébastien Hulot; Samreen Ijaz; Hajnal-Gabriela Illes; Patrick Imbert; Hugo Inácio; Elias Iosifidis; Sarah Isgett; Palliya Guruge Pramodya Ishani Ishani; Tiago Isidoro; Margaux Isnard; Junji Itai; Daniel Ivulich; Danielle Jaafar; Salma Jaafoura; Julien Jabot; Clare Jackson; Nina Jamieson; Pierre Jaquet; Coline Jaud-Fischer; Stéphane Jaureguiberry; Issrah Jawad; Florence Jego; Synne Jenum; Ruth Jimbo-Sotomayor; N Ruth; Jorge García; Cédric Joseph; Mark Joseph; Swosti Joshi; Mercé Jourdain; Philippe Jouvet; Jennifer June; Anna Jung; Hanna Jung; Dafsah Juzar; Ouifiya Kafif; Florentia Kaguelidou; Sabina Kali; Smaragdi Kalomoiri; Darshana Hewa Kandamby; Chris Kandel; Ravi Kant; Dyah Kanyawati; Christiana Kartsonaki; Daisuke Kasugai; Anant Kataria; Kevin Katz; Simreen Kaur Johal; Christy Kay; Hannah Keane; Seán Keating; Andrea Kelly; Aoife Kelly; Claire Kelly; Niamh Kelly; Sadie Kelly; Yvelynne Kelly; Maeve Kelsey; Ryan Kennedy; Kalynn Kennon; Maeve Kernan; Younes Kerroumi; Sharma Keshav; Evelyne Kestelyn; Imrana Khalid; Antoine Khalil; Coralie Khan; Irfan Khan; Michelle E Kho; Saye Khoo; Yuri Kida; Peter Kiiza; Anders Benjamin Kildal; Jae Burm Kim; Antoine Kimmoun; Detlef Kindgen-Milles; Alexander King; Nobuya Kitamura; Paul Klenerman; Gry Kloumann Bekken; Stephen Knight; Robin Kobbe; Chamira Kodippily; Malte Kohns Vasconcelos; Volkan Korten; Caroline Kosgei; Arsène Kpangon; Karolina Krawczyk; Sudhir Krishnan; Oksana Kruglova; Deepali Kumar; Bharath Kumar Tirupakuzhi Vijayaraghavan; Pavan Kumar Vecham; Ethan Kurtzman; Neurinda Permata Kusumastuti; Demetrios Kutsogiannis; Galyna Kutsyna; Konstantinos Kyriakoulis; Marie Lachatre; Marie Lacoste; John G Laffey; Marie Lagrange; Fabrice Laine; Olivier Lairez; Antonio Lalueza; Marc Lambert; François Lamontagne; Marie Langelot-Richard; Vincent Langlois; Eka Yudha Lantang; Marina Lanza; Cédric Laouénan; Samira Laribi; Delphine Lariviere; Stéphane Lasry; Odile Launay; Didier Laureillard; Yoan Lavie-Badie; Andrew Law; Cassie Lawrence; Teresa Lawrence; Minh Le; Clément Le Bihan; Cyril Le Bris; Georges Le Falher; Lucie Le Fevre; Quentin Le Hingrat; Marion Le Maréchal; Soizic Le Mestre; Gwenaël Le Moal; Vincent Le Moing; Paul Le Turnier Hervé Le Nagard; Ema Leal; Marta Leal Santos; James Lee; Su Hwan Lee; Todd C Lee; Gary Leeming; Bénédicte Lefebvre; Laurent Lefebvre; Benjamin Lefevre; Sylvie Le Gac; Jean-Daniel Lelievre; François Lellouche; Adrien Lemaignen; Véronique Lemee; Anthony Lemeur; Gretchen Lemmink; Rafael León; Marc Leone; Michela Leone; François-Xavier Lescure; Olivier Lesens; Mathieu Lesouhaitier; Amy Lester-Grant; Bruno Levy; Yves Levy;

Claire Levy-Marchal; Erwan L'Her; Gianluigi Li Bassi; Janet Liang; Geoffrey Liegeon; Wei Shen Lim; Chantre Lima; Bruno Lina; Andreas Lind; Guillaume Lingas; Sylvie Lion-Daolio; Keibun Liu; Marine Livrozet; Patricia Lizotte; Antonio Loforte; Navy Lolong; Diogo Lopes; Dalia Lopez-Colon; Anthony L Loschner; Paul Loubet; Bouchra Loufti; Guillame Louis; Silvia Lourenco; Jean Christophe Lucet; Carlos Lumbreras Bermejo; Carlos M Luna; Olguta Lungu; Liem Luong; Nestor Luque; Dominique Luton; Ruth Lyons; Olavi Maasikas; Oryane Mabiala; Samual Mac Donald; Moïse Machado; Gabriel Macheda; Juan Macias Sanchez; Jai Madhok; Hashmi Madiha; Guillermo Maestro de la Calle; Rafael Mahieu; Sophie Mahy; Ana Raquel Maia; Lars S Maier; Mylène Maillet; Thomas Maitre; Maximilian Malfertheiner; Nadia Malik; Paddy Mallon; Fernando Maltez; Denis Malvy; Victoria Manda; Jose M Mandei; Laurent Mandelbrot; Frank Manetta; Julie Mankikian; Edmund Manning; Aldric Manuel; Ceila Maria Sant'Ana Málaque; Daniel Marino; Flávio Marino; Samuel Markowicz; Ana Marques; Catherine Marquis; Brian Marsh; Laura Marsh; Megan Marshal; John Marshall; Celina Turchi Martelli; Emily Martin; Guillaume Martin-Blondel; Alessandra Martinelli; Ignacio Martin-Loeches; Martin Martinot; Ana Martins; João Martins; Nuno Martins; Caroline Martins Rego; Gennaro Martucci; Olga Martynenko; Eva Miranda Marwali; Juan Fernado Masa Jimenez; David Maslove; David Maslove; Sabina Mason; Basri Mat Nor; Moshe Matan; Daniel Mathieu; Mathieu Mattei; Romans Matulevics; Laurence Maulin; Michael Maxwell; Javier Maynar; Thierry Mazzoni; Natalie Mc Evoy; Lisa Mc Sweeney; Lorraine McAndrew; Colin McArthur; Aine McCarthy; Anne McCarthy; Colin McCloskey; Rachael McConnochie; Sherry McDermott; Sarah E McDonald; Aine McElroy; Samuel McElwee; Victoria McEneany; Allison McGeer; Chris McKay; Johnny McKeown; Kenneth A McLean; Bairbre McNicholas; Elaine McPartlan; Edel Meaney; Cécile Mear-Passard; Maggie Mechlin; Omar Mehkri; Ferruccio Mele; Luis Melo; Joao Joao Mendes; Ogechukwu Menkiti; Kusum Menon; France Mentré; Alexander J Mentzer; Emmanuelle Mercier; Noémie Mercier; Antoine Merckx; Mayka Mergeay-Fabre; Blake Mergler; Laura Merson; António Mesquita; Osama Metwally; Agnès Meybeck; Dan Meyer; Alison M Meynert; Vanina Meysonnier; Amina Meziane; Mehdi Mezidi; Céline Michelanglei; Isabelle Michelet; Efstathia Mihelis; Vladislav Mihnovit; Hugo Miranda-Maldonado; Asma Moin; David Molina; Elena Molinos; Brenda Molloy; Mary Mone; Agostinho Monteiro; Claudia Montes; Giorgia Montrucchio; Sarah Moore; Shona C Moore; Lina Morales Cely; Lucia Moro; Catherine Motherway; Ana Motos; Hugo Mouquet; Clara Mouton Perrot; Julien Moyet; Jimmy Mullaert; Fredrik Muller; Karl Erik Müller; Aisling Murphy; Aisling Murphy; Lorna Murphy; Marlène Murris; Srinivas Murthy; Himed Musaab; Dimitra Melia Myrodia; Dave Nagpal; Alex Nagrebetsky; Mangala Narasimhan; Nadège Neant; Holger Neb; Raul Neto; Emily Neumann; Bernardo Neves; Pauline Yeung Ng; Anthony Nghi; Duc Nguyen; Orna Ni Choileain; Niamh Ni Leathlobhair; Alistair Nichol; Prompak Nitayavardhana; Stephanie Nonas; Marion Noret; Lisa Norman; Alessandra Notari; Mahdad Noursadeghi; Karolina Nowicka; Saad Nseir; Jose I Nunez; Nurnaningsih Nurnaningsih; Elsa Nyamankolly; Fionnuala O Brien; Annmarie O Callaghan; Annmarie O'Callaghan; Giovanna Occhipinti; Derbrenn OConnor; Max O'Donnell; Tawnya Ogston; Takayuki Ogura; Tak-Hyuk Oh; Sophie O'Halloran; Katie O'Hearn; Shinichiro Ohshimo; Agnieszka Oldakowska; João Oliveira; Larissa Oliveira; Piero L Olliaro; David SY Ong; Wilna Oosthuyzen; Anne Opavsky; Peter Openshaw; Claudia Milena Orozco-Chamorro; Jamel Ortoleva; Javier Osatnik; Linda O'Shea; Miriam O'Sullivan; Nadia Ouamara; Rachida Ouissa; Clark Owyang; Eric Oziol; HM Upulee Pabasara; Justine Pages Maïder Pagadoy; Amanda Palacios; Mario Palacios; Massimo Palmarini; Giovanna Panarello; Prasan Kumar Panda; Mauro Panigada; Nathalie Pansu; Aurélie Papadopoulos; Melissa Parker; Briseida Parra; Jérémie Pasquier; Bruno Pastene; Fabian Patauner; Luís Patrão; Patricia Patricio; Juliette Patrier; Lisa Patterson; Christelle Paul; Mical Paul; Jorge Paulos; William A Paxton; Jean-François Payen; Miguel Pedrera Jiménez; Giles J Peek; Florent Peelman; Nathan Peiffer-Smadja; Vincent Peigne; Mare Pejkovska; Paolo Pelosi; Ithan D Peltan; Rui Pereira; Daniel Perez; Luis Periel; Thomas Perpoint; Antonio Pesenti; Vincent Pestre; Lenka Petrou; Ventzislava Petrov-Sanchez; Frank Olav Pettersen; Gilles Peytavin; Scott Pharand; Michael Piagnerelli; Walter Picard; Olivier Picone; Mariade Piero; Carola Pierobon; Carlos Pimentel; Raquel Pinto; Catarina Pires; Isabelle Pironneau; Lionel Piroth; Riinu Pius; Simone Piva; Laurent Plantier; Daniel Plotkin; Julien Poissy; Ryadh Pokeerbux; Maria Pokorska-Spiewak; Sergio Poli; Georgios Pollakis; Diane Ponscarme; Jolanta Popielska; Andra-Maris Post; Douwe F Postma; Pedro Povoa; Diana Póvoas; Jeff Powis; Sofia Prapa; Sébastien Preau; Christian Prebensen; Jean-Charles Preiser; Anton Prinssen; Mark G Pritchard; Gamage Dona Dilanthi Priyadarshani;

Lucia Proença; Sravya Pudota; Oriane Puéchal; Bambang Pujo Semedi; Gregory Purcell; Luisa Quesada; Vilmaris Quinones-Cardona; Víctor Quirós González; Else Quist-Paulsen; Mohammed Quraishi; Christian Rabaud; Aldo Rafael; Marie Rafiq; Fernando Rainieri; Nagarajan Ramakrishnan; Blandine Rammaert; Ritika Ranjan; Christophe Rapp; Aasiyah Rashan; Thalha Rashan; Menaldi Rasmin; Indrek Rätsep; Cornelius Rau; Ali Raza; Andre Real; Stanislas Rebaudet; Sarah Redl; Brenda Reeve; Liadain Reid; Liadain Reid; Dag Henrik Reikvam; Renato Reis; Jordi Rello; Jonathan Remppis; Martine Remy; Hongru Ren; Hanna Renk; Liliana Resende; Anne-Sophie Resseguier; Matthieu Revest; Oleksa Rewa; Luis Felipe Reyes; Tiago Reyes; Maria Ines Ribeiro; David Richardson; Denise Richardson; Laurent Richier; Jordi Riera; Ana L Rios; Asgar Rishu; Patrick Rispal; Karine Risso; Nicholas Rizer; Chiara Robba; André Roberto; Stephanie Roberts; David L Robertson; Olivier Robineau; Ferran Roche-Campo; Paola Rodari; Simão Rodeia; Julia Rodriguez Abreu; Bernhard Roessler; Pierre-Marie Roger; Emmanuel Roilideseira; Amanda Rojek; Juliette Romaru; Roberto Roncon-Albuquerque; Mélanie Roriz; Manuel Rosa-Calatrava; Michael Rose; Dorothea Rosenberger; Andrea Rossanese; Matteo Rossetti; Bénédicte Rossignol; Patrick Rossignol; Stella Rousset; Carine Roy; Benoît Roze; Desy Rusmawatiningtyas; Clark D Russell; Maeve Ryan; Maria Ryan; Steffi Ryckaert; Aleksander Rygh Holten; Isabela Saba; Musharaf Sadat; Valla Sahraei; Nadia Saidani; Maximilien Saint-Gilles; Pranya Sakiyalak; Leonardo Salazar; Gabriele Sales; Stéphane Sallaberry; Charlotte Salmon Gandonniere; Hélène Salvator; Olivier Sanchez; Angel Sanchez-Miralles; Vanessa Sancho-Shimizu; Gyan Sandhu; Zulfiqar Sandhu; Pierre-François Sandrine; Oana Sandulescu; Marlene Santos; Shirley Sarfo-Mensah; Bruno Sarmento Banheiro; Iam Claire E Sarmiento; Benjamine Sarton; Sree Satyapriya; Rumaisah Satyawati; Egle Saviciute; Parthena Savvidou; Justin Schaffer; Tjard Schermer; Arnaud Scherpereel; Marion Schneider; Stephan Schroll; Michael Schwameis; Janet T Scott; James Scott-Brown; Nicholas Sedillot; Tamara Seitz; Caroline Semaille; Malcolm G Semple; Eric Senneville; Claudia Sepulveda; Filipa Sequeira; Tânia Sequeira; Pablo Serrano Balazote; Ellen Shadowitz; Mohammad Shamsah; Shaikh Sharjeel; Pratima Sharma; Catherine A Shaw; Victoria Shaw; Haixia Shi; Mohiuddin Shiekh; Takuya Shiga; Nobuaki Shime; Hiroaki Shimizu; Keiki Shimizu; Naoki Shimizu; Sally Shrapnel; Hoi Ping Shum; Nassima Si Mohammed; Jeanne Sibiude; Atif Siddiqui; Louise Sigfrid; Piret Sillaots; Catarina Silva; Maria Joao Silva; Rogério Silva; Wai Ching Sin; Budha Charan Singh; Punam Singh; Pompini Agustina Sitompul; Vegard Skogen; Sue Smith; Benjamin Smood; Coilin Smyth; Michelle Smyth; Michelle Smyth; Morgane Snacken; Dominic So; Monserrat Solis; Joshua Solomon; Tom Solomon; Emily Somers; Agnès Sommet; Myung Jin Song; Rima Song; Tae Song; Jack Song Chia; Michael Sonntagbauer; Alberto Sotto; Edouard Soum; Ana Chora Sousa; Marta Sousa; Maria Sousa Uva; Vicente Souza-Dantas; Alexandra Sperry; BP Sanka Ruwan Sri Darshana; Shiranee Sriskandan; Sarah Stabler; Thomas Staudinger; Stephanie-Susanne Stecher; Ymkje Stienstra; Birgitte Stiksrud; Eva Stolz; Amy Stone; Adrian Streinu-Cercel; Anca Streinu-Cercel; Samantha Strudwick; Ami Stuart; David Stuart; Gabriel Suen; Jacky Y Suen; Prasanth Sukumar; Asfia Sultana; Charlotte Summers; Dubravka Supic; Magdalena Surovcová; Konstantinos Syrigos; Jaques Sztajnbok; Konstanty Szuldrzynski; Shirin Tabrizi; Fabio S Taccone; Lysa Tagherset; Ewa Talarek; Sara Taleb; Jelmer Talsma; Maria Lawrensia Tampubolon; Le Van Tan; Hiroyuki Tanaka; Taku Tanaka; Hayato Taniguchi; Coralie Tardivon; Pierre Tattevin; MAzhari Taufik; Hassan Tawfik; Richard S Tedder; João Teixeira; Sofia Tejada; Marie-Capucine Tellier; Vanessa Teotonio; François Téoulé; Pleun Terpstra; Olivier Terrier; Nicolas Terzi; Hubert Tessier-Grenier; Adrian Tey; Vincent Thibault; Simon-Djamel Thiberville; Benoît Thill; Shaun Thompson; David Thomson; Emma C Thomson; Duong Bich Thuy; Ryan S Thwaites; Paul Tierney; Vadim Tieroshyn; Jean-François Timsit; Noémie Tissot; Maria Toki; Timo Tolppa; Kristian Tonby; Antoni Torres; Margarida Torres; Hernando Torres-Zevallos; Michael Towers; Théo Treoux; Huynh Trung Trieu; Cécile Tromeur; Ioannis Trontzas; Tiffany Trouillon; Jeanne Truong; Christelle Tual; Sarah Tubiana; Helen Tuite; Jean-Marie Turmel; Lance CW Turtle; Pawel Twardowski; Makoto Uchiyama; PGIshara Udayanga; Roman Ullrich; Alberto Uribe; Asad Usman; Cristinava Vajdovics; Luís Val-Flores; Ana Luiza Valle; Amélie Valran; Stijn Van de Velde; Marcelvan den Berge; Machteld Van der Feltz; Nicky Van Der Vekens; Peter Van der Voort; Sylvie Van Der Werf; Lauravan Gulik; Jarne Van Hattem; Stevenvan Lelyveld; Carolienvan Netten; Gitte Van Twillert; Noémie Vanel; Henk Vanoverschelde; Pooja Varghese; Michael Varrone; Charline Vauchy; Helen Vaughan; Aurélie Veislinger; Sebastian Vencken; Sara Ventura; Annelies Verbon; José Ernesto Vidal; César Vieira; Joy Ann Villanueva; Judit Villar; Pierre-Marc Villeneuve; Andrea

Villoldo; Nguyen Van Vinh Chau; Benoit Visseaux; Hannah Visser; Chiara Vitiello; Fanny Vuotto; Chih-Hsien Wang; Jia Wei; Katharina Weil; Sanne Wesselius; Murray Wham; Bryan Whelan; Nicole White; Paul Henri Wicky; Aurélie Wiedemann; Surya Otto Wijaya; Keith Wille; Virginie Williams; Evert-Jan Wils; Ng Wing Yiu; Calvin Wong; Ioannis Xynogalas; Sophie Yacoub; Masaki Yamazaki; Yazdan Yazdanpanah; Cécile Yelnik; Stephanie Yerkovich; Toshiki Yokoyama; Hodane Yonis; Obada Yousif; Saptadi Yuliarto; Akram Zaaqoq; Marion Zabbe; Kai Zacharowski; Maram Zahran; Maria Zambon; Miguel Zambrano; Alberto Zanella; Konrad Zawadka; Hiba Zayyad; Alexander Zoufaly; David Zucman

## Funding

| Funder | Grant reference number | Author |
|---|---|---|
| Wellcome Trust | 215091/Z/18/Z | Joaquín Baruch<br>Jake Dunning<br>Martina Escher<br>Jia Wei<br>Peter W Horby<br>Piero Luigi Olliaro |
| Bill and Melinda Gates Foundation | OPP1209135 | Joaquín Baruch<br>Jake Dunning<br>Martina Escher<br>Jia Wei<br>Peter W Horby<br>Piero Luigi Olliaro |
| University of Oxford's COVID-19 Research Response Fund | 0009146 | Laura Merson |
| H2020 European Research Council | 101003589 | ISARIC Clinical Characterisation Group |
| European Clinical Research Alliance on Infectious Diseases (ECRAID) | 965313 | ISARIC Clinical Characterisation Group |
| Health Research Board of Ireland | CTN-2014–12 | ISARIC Clinical Characterisation Group |
| National Institute for Health Research | CO-CIN-01 | ISARIC Clinical Characterisation Group |
| Medical Research Council | MC_PC_19059 | ISARIC Clinical Characterisation Group |
| Public Health England | 200907 | ISARIC Clinical Characterisation Group |
| Wellcome Trust | 205228/Z/16/Z | ISARIC Clinical Characterisation Group |
| Respiratory Infections at Imperial College London with PHE | 200927 | ISARIC Clinical Characterisation Group |
| NIHR Biomedical Research Centre at Imperial College London | IS-BRC1215–20013 | ISARIC Clinical Characterisation Group |
| Research Council of Norway | 312780 | ISARIC Clinical Characterisation Group |
| Liverpool Experimental Cancer Medicine Centre | C18616/A25153 | ISARIC Clinical Characterisation Group |
| CIHR Coronavirus Rapid Research Funding Opportunity | OV2170359 | ISARIC Clinical Characterisation Group |

The funders had no role in study design, data collection and interpretation, or the decision to submit the work for publication.

## Author contributions

ISARIC Clinical Characterisation Group, Funding acquisition; Matthew D Hall, Conceptualization, Formal analysis, Methodology, Software, Visualization, Writing – original draft, Writing – review and editing; Joaquín Baruch, Andrew Dagens, Writing – original draft, Writing – review and editing; Gail Carson, Jake Dunning, Amanda Rojek, Conceptualization, Writing – original draft, Writing – review and editing; Barbara Wanjiru Citarella, Data curation, Project administration, Writing – review and editing; Emmanuelle A Dankwa, Mark Pritchard, Formal analysis, Software, Writing – review and editing; Christl A Donnelly, Methodology, Writing – review and editing; Martina Escher, Jia Wei, Writing – review and editing; Christiana Kartsonaki, Formal analysis, Methodology, Software, Writing – original draft, Writing – review and editing; Laura Merson, Conceptualization, Funding acquisition, Project administration, Writing – review and editing; Peter W Horby, Conceptualization, Writing – review and editing; Piero L Olliaro, Conceptualization, Funding acquisition, Writing – original draft, Writing – review and editing

## Author ORCIDs

Matthew D Hall http://orcid.org/0000-0002-2671-3864
Laura Merson http://orcid.org/0000-0002-4168-1960

## Ethics

Human subjects: The study was approved by the World Health Organization Ethics Review Committee (RPC571 and RPC572). Local ethics approval was obtained for each participating country and site according to local requirements. All necessary patient/participant consent has been obtained and the appropriate institutional forms have been archived. All necessary patient/participant consent has been obtained and the appropriate institutional forms have been archived.

## Decision letter and Author response

Decision letter https://doi.org/10.7554/eLife.70970.sa1
Author response https://doi.org/10.7554/eLife.70970.sa2

---

# Additional files

## Supplementary files

- Supplementary file 1. Description of variables used in regression analyses.

- Supplementary file 2. Extended demographics table for the complete dataset, by month of COVID-19 admission and overall. Numbers are raw counts with column-wise percentages in brackets.

- Supplementary file 3. Prevalence of symptoms at admission amongst individuals with no reported "common" symptoms (cough, fatigue, fever or shortness of breath), by age group. Numbers are percentages with fractions in brackets.

- Supplementary file 4. Full results of the multivariable linear regression analysis identifying correlates of time from symptom onset to hospital admission (question 1). Given are percentage predicted increased in time to admission (in days), 95 % confidence intervals, and the *p*-values of Wald tests for the inclusion of each variable in each regression.

- Supplementary file 5. Full results of the multivariable logistic regression analysis identifying predictors of ICU/HDU admission (question 2). The *p*-values of Wald tests for the inclusion of each variable in each regression are included as a separate column. Given are odds ratios for admission, 95 % confidence intervals, and the *p*-values of Wald tests for the inclusion of each variable in each regression.

- Supplementary file 6. Full results of the multivariable logistic regression analysis identifying correlates of time to ICU/HDU admission amongst patients so admitted (question 3). Given are percentage predicted increased in time to admission (in days), 95 % confidence intervals, and the *p*-values of Wald tests for the inclusion of each variable in each regression.

- Supplementary file 7. Extended version of *Table 4*, including the country variable and the "unknown" class for comorbidities.

- Supplementary file 8. Full details for the ISARIC Clinical Characterisation Group.

- Transparent reporting form

• Reporting standard 1. Strobe statement.

## Data availability

The data that underpin this analysis are highly detailed clinical data on individuals hospitalised with COVID-19. Due to the sensitive nature of these data and the associated privacy concerns, they are available via a governed data access mechanism following review of a data access committee. Data can be requested via the IDDO COVID-19 Data Sharing Platform (http://www.iddo.org/covid-19). The Data Access Application, Terms of Access and details of the Data Access Committee are available on the website. Briefly, the requirements for access are a request from a qualified researcher working with a legal entity who have a health and/or research remit; a scientifically valid reason for data access which adheres to appropriate ethical principles. The full terms are at https://www.iddo.org/document/covid-19-data-access-guidelines. A small subset of sites who contributed data to this analysis have not agreed to pooled data sharing as above. In the case of requiring access to these data, please contact the corresponding author in the first instance who will look to facilitate access. We have provided the R code used to process data and run the regression analysis at https://github.com/ISARICDataPlatform/InpatientJourneyDataProcessing, (copy archived at swh:1:rev:ce42035d-6cf80852089d95264215f7bb487cb998). Source data for all figures has also been provided.

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
