## [Editor Report]

This large multicenter study tracked the clinical journeys of COVID-19 hospitalized patients over 2020, and found variations in clinical outcomes over time. This paper will be of interest to the large class of clinicians, public health workers and health policy makers who want to know the variation in the nature and duration of the care provided to hospitalised patients during an infectious disease epidemic. The study highlights the importance of maintaining the capacity of registration of infectious disease like COVID-19, during a pandemic and after. While the cohort recruited patients from multiple countries, the vast majority of patients came from the UK, so the results are most applicable to this country.

---

## [Decision Letter]

**Decision letter after peer review:**

Thank you for submitting your article "Ten months of temporal variation in the clinical journey of hospitalised patients with COVID-19: an observational cohort" for consideration by *eLife*. Your article has been reviewed by 2 peer reviewers, one of whom is a member of our Board of Reviewing Editors, and the evaluation has been overseen by Miles Davenport as the Senior Editor. The reviewers have opted to remain anonymous.

Essential revisions:

1. Provide further rationale for the separation of gastrointestinal and common symptoms for analyses.

2. There are several indications that the cohort may not be entirely representative of all hospitalised patients with COVID-19 in the study settings, including a surprisingly high case fatality rate and few participants recruited at some study sites. Please provide more discussion regarding the representativeness of the cohort, eligibility criteria, and study participation rates.

3. Please provide date of end of follow-up and an estimate of the average/total follow-up time.

4. Indicate whether all participants were followed-up over equivalent periods of time to assess study outcomes, in order to justify the use of logistic regression rather than survival analysis.

5. Provide more information regarding what variables are being included in the model, and whether Table 4 represents results from a model including all variables or multiple separate models. This influences how readers interpret the variable coefficients, as it is important to know what variables have been adjusted for and whether the ratios should be interpreted as crude or adjusted associations, and whether interactions are being accounted for in the estimated effects.

6. While less essential, the reviewers made several other suggestions/queries that should be considered by the authors in their revision.

*Reviewer #1:*

In this study, COVID-19 hospitalized patient trajectories were tracked over 2020 in order to assess whether patient outcomes changed over time. The study of these time trends can be useful for assessing how health systems respond to pandemics, and may help in better planning for future outbreaks and pandemics.

Some of the major strengths of this study include its very large sample size, which allowed a detailed examination of clinical pathways of patients, and the ability to demonstrate clear time trends by age group, as well as interesting interactions between age and ICU admission on the risk of death. The study was able to look at multiple outcomes of interest, including death, discharge, ICU admission, and time to these events. Detailed analyses are provided in either the main text or the appendix.

As the authors point out, it is likely that temporal changes in patient outcomes were influenced not only by changes in clinical management and treatments over the course of the pandemic, but potentially by surges in the numbers of admissions which may have taxed the system capacity and quality of care at specific points in time. The authors rightly point out that system capacity should be adequately controlled for when assessing the effects of interventions against COVID-19. However, in their own analysis they do not control for any variable that would be an indicator of system capacity, and do not provide any suggestions for how other studies could control for this. Therefore, the analysis in this study by itself does not demonstrate that system capacity had an impact on patient outcomes. The reasons underlying the time trends they observe therefore remain unclear.

While the cohort recruited patients from multiple countries, the vast majority (83%) of patients came from the UK. It is therefore likely that most time trends reflect the evolution of the pandemic and health-seeking patterns in the UK. It is not clear to what extent the time trends are generalizable to other countries who may have had epidemic peaks at different times and that may have different holidays (which are deeply influential on infection patterns and health-seeking behaviors).

It is unclear to what extent selection bias may have influenced the results and observed time trends. In terms of selection into the cohort, there is not much information on how study sites were selected, when they started contributing cases to the cohort, if there were any additional selection criteria applied to participants beyond being a confirmed or suspected COVID-19 case, and what proportion of theoretically eligible participants were recruited into the study. The low number of participants recruited at some study sites suggests that the recruited participants only represent a fraction of all potentially eligible COVID-19 patients at each study site. This may have led to a selection bias if those who were not recruited differed in some systematic way from those included. If recruitment probability varied over time, then this might have also influenced the time trends observed. In terms of selection out of the cohort, there is little information on how many patients were lost to follow-up, had missing outcome data, or were transferred to other facilities. It is possible these patients may have systematic differences from those with complete data, which may lead to selection bias.

Comments for the authors:

1. Please provide date of end of follow-up and an estimate of the average/total follow-up time (STROBE item 14C).

2. Please provide more information regarding participant recruitment, such as numbers potentially eligible and recruited into the study (STROBE item 5,6,13). Were all eligible COVID-19 cases included, or were only some included? Are there more eligibility criteria than those mentioned? The discussion suggests it is the latter (which the authors refer to as "enrolment bias", would recommend rephrasing this as selection bias). If site case loads influenced recruitment then this could potentially lead to selection bias if certain types of patients are favored for study inclusion and this varies over time.

3. Please report in the main text the number (%) of patients lost to follow-up or with missing data for each analysis (STROBE item 13,14).

4. It is unusual to have an outcome of odds or probability of an event where the time frame over which the patient can experience the outcome is not specified. Presumably the outcomes for question 2 and 4 are ICU admission or death over the entire study period. This is not a fair comparison, because patients recruited in later months may not have been followed-up as long as patients recruited earlier, and the follow-up of patients may have varied over the duration of the study. It is generally more rigorous to specify a time frame over which the outcome is assessed and only analyze patients with that follow-up time (ex. death within 3 months of admission) or to use analysis methods which account for differential follow-up time of participants, such as Kaplan-Meier or survival analyses. I would suggest modifying the logistic regressions to have the outcome over a specified time frame or to use survival analyses instead (or to explain why these methods were not used).

5. P39: "Patients lost to follow-up before any of these outcomes were included unless the time to that event was the outcome of interest" It is unclear to me what this means. Does this mean that patients lost to follow-up are included as non-events in the denominator?? It seems to me a better way to treat these patients would be to exclude them or better yet to include them as right-censored observations in survival analyses, as suggested above.

6. Table 4:

– There is an excessive number of decimal places in this table (pseudo-precision) that are not clinically meaningful. I would suggest rounding all values to the nearest decimal (odds ratios) or to the nearest percentage point (percentages). Reducing the number of decimal points will make this table much simpler to read and understandable.

– Please consider dropping the p-values for each individual value. The confidence intervals are much more informative as they indicate the precision around each estimate, convey the same information, and put the emphasis on the estimation of the parameter rather than its statistical significance. The small p-values in most cases are a reflection of the large sample size.

– Conversely, the addition of p-values for the Wald type III test for the overall effect of each variable to the table would be informative.

– The above suggestions (decreasing decimal places and dropping p-values) should create more space in the table. Please consider using this space to add descriptive data on the probability of experiencing death and the number days to death or discharge for each category. This descriptive information helps to contextualize the odds ratios and relative time increases, and may be more clinically useful for some readers than the ratio measures.

– It is not clear whether the results from this table present univariate or multivariable regression model results (presumably the latter). If the analyses are multivariable, then all the variables that were included in the model should be described in a footnote, as this influences the interpretation of the results. For example, it is important to know whether the monthly estimates are adjusted for sex and age, as the age and sex of patients varied over time.

– The pregnancy variable is conflated with the effect of sex because presumably all the men where categorized as 0 for this variable. This variable should be treated as an interaction variable with sex (sex + sex*pregnancy) in order to separate the effect of sex from the effect of pregnancy.

7. There is no Table 3.

8. The authors mention system capacity during epidemic surges as a potential contributor to time trends, but did not include any indicator of system capacity in their analysis, so this interpretation remains speculative. Have the authors considered included such indicators as a variable in their multivariable model (ex. number of COVID-19 infections or hospitalizations reported during the day/week of admission in each country)?

*Reviewer #2:*

This paper reports here the multi-national collaborative effort on establishment of cohort of 142,540 patients hospitalised with COVID-19. The authors confirmed the previous findings of the trend of hospitalization, ICU admission and case fatality ratio. The author not only shared the large cohort data, including UK and other countries, they also did Multivariable logistic or linear regression to investigate factors associated with the main outcomes, including time from symptom onset to hospital admission, probability of ICU/HDU admission, time from hospital admission to ICU/HDU admission, case fatality ratio (CFR) and total length of hospital stay.

Strengths:

This is a large database from ISARIC Clinical Characterisation Group. Based on the large data, the author answered six important questions, especially time to hospital admission, ICU admission, length of stay and case fatality ratio. The findings are valuable, such as health systems may at times be overwhelmed; system capacity may be an important predictor of patient outcome and may supersede other factors such as increasing case management skills and the influence of new therapies.

Weaknesses:

Although the paper does have strengths in principle, the weaknesses of the paper are that these strengths are not directly demonstrated. In particular:

1. In this study, why gastrointestinal symptoms (abdominal pain, diarrhoea, and vomiting) are emphasized?

2. In this study, patients with a recorded hospital admission date before their symptom onset date, were taken to be a nosocomial infection. But patients who had got infection after exposure to virus may present symptoms later after admission. That is ,on the day of admission, the patient may be in the incubation period.

3. Similarly, "patients with nosocomial infections had lower odds of being admitted to ICU/HDU (OR 0.68, 95% CI 0.62-255 0.74)". The underlying reason may be not nosocomial infection, but that these patients had been admitted earlier than those with adverse outcomes.

4. Line 205: "times" should be "time". Similarly, in the lines 209 and 218.

5. For admission to ICU as an example, the comparison between different months means little, as there are many confounding factors for admission to ICU. One and half years have passed, it is time to look forward to improve treatment.

6. Why less elderly patients were admitted to ICU?

7. Why a wide variety of serious or chronic medical conditions were associated with lower odds of ICU admission. Is it because of lower admission of elderly patients?

8. The raw case fatality rate (CFR) was extremely high in this paper. The readers would be interested in the representativeness of this cohort.

9. In this study, the authors confirmed what is known that Age, male and comorbidities are risk for higher mortality. The readers would like to understand why asthma was associated with lower mortality.

10. Discussion: "patients admitted after experiencing symptoms for longer than 1 week were less likely to die." I am afraid that the explanation lacks scientific evidence.

[Editors' note: further revisions were suggested prior to acceptance, as described below.]

Thank you for resubmitting your work entitled "Ten months of temporal variation in the clinical journey of hospitalised patients with COVID-19: an observational cohort" for further consideration by *eLife*. Your revised article has been evaluated by Miles Davenport as the Senior Editor, and a Reviewing Editor.

The manuscript has been improved but there are some remaining issues that need to be addressed, as outlined below:

1. The authors mention that the time periods over which the probability of ICU admission and death/discharge were measured are implicit in the cutoff times for time to event analyses (13 days and 45 days). This should be made explicit in the manuscript by indicating that the outcomes were ICU admission within 13 days for question 2 and death/discharge within 45 days for question 4.

2. While the authors specify that all analyses are multivariable in the text, this should also be mentioned in the table legends and/or footnotes in order for tables to be self-sufficient.

3. Table 1: because 'nosocomial infection' is no longer defined in the methods, this row should be relabeled as post-admission symptom onset infections

4. There is an error in Table 2, the rows for symptoms at admission are mistakenly labeled as comorbidities and vice-versa.

5. The authors refer to an non-existent supplementary figure 2 in the results. Presumably they mean Figure 2. This should be fixed.

6. Figure 4 is missing a C inset.

7. On line 446 the authors refer to Figure 3 when they should be referring to Figure 4.

8. Table 3 appears to be lacking the standard deviation for time from symptom onset to hospital admission.

---

## [Author Response]

Essential revisions:1. Provide further rationale for the separation of gastrointestinal and common symptoms for analyses.

After consideration, we have removed the gastrointestinal symptoms variable from the regressions. They were a legacy of earlier iterations of this work, and we agree with the reviewer’s implicit query about what they really add. Its removal makes little appreciable difference to the results. The common symptoms score is only used for question 1, and the main observation regarding it is the initially counter-intuitive finding that patients with fewer symptoms are admitted more quickly, which we explain in the discussion (L984-996).

2. There are several indications that the cohort may not be entirely representative of all hospitalised patients with COVID-19 in the study settings, including a surprisingly high case fatality rate and few participants recruited at some study sites. Please provide more discussion regarding the representativeness of the cohort, eligibility criteria, and study participation rates.

Our reported CFR is in line with that from Docherty et al. (http://doi.org/10.1016/S2213-2600(21)00175-2), and, with UK patients from early in the epidemic forming a very large proportion of our total, as well as the elderly profile of this patient population, this is not surprising. We acknowledge that our recruitment will be influenced by the capacity of each site to enrol patients at any given time and to complete their follow-up. However, there are no obvious reason why either enrolment of completion of follow-up should be biased with respect to our outcomes. Patients were prospectively enrolled, at which point all outcomes would be unknown, and final outcomes would be unknown at the point where follow-up is abandoned. We have expanded this part of the limitations paragraph (L1094-1103).

On a related point, we have specified throughout that the case fatality rate is the hospital CFR (hCFR).

3. Please provide date of end of follow-up and an estimate of the average/total follow-up time.

The last date at which it was possible that any patients in this set to receive follow-up was 8 March, some time after any patients with a recorded outcome had died or been discharged. We have added this to the text (L85-86, 517-521). We have also added median and IQR follow-up times for both all patients and patients with censored outcomes (L615-617).

4. Indicate whether all participants were followed-up over equivalent periods of time to assess study outcomes, in order to justify the use of logistic regression rather than survival analysis.

While it was not explicit in the text of the initial submission, these analyses do in fact look at ICU admission within 13 days (question 3) or death or discharge within 45 days (question 5), because we omitted those with longer time periods as having possibly mis-entered data. Patients were also able to be followed up more than 45 days into 2021, so there is no issue with censorship at the end of the study period. This has been clarified in the text (L595-613). The limit for the time to admission analysis was 24 days, but as all patients here were admitted, there is no censorship issue in this case.

On the topic of survival analysis, this was indeed our initial approach. However, as there were two outcomes (death and discharge), this approach required a competing risks model, and we found that treating those two outcomes as independent events that could occur at any time led to CFR estimates that differed considerably to those calculated using a simple stratified analysis of risk groups. We felt that this was due to invalid assumptions about how the outcomes occur in reality, and opted to pursue the current approach instead.

5. Provide more information regarding what variables are being included in the model, and whether Table 4 represents results from a model including all variables or multiple separate models. This influences how readers interpret the variable coefficients, as it is important to know what variables have been adjusted for and whether the ratios should be interpreted as crude or adjusted associations, and whether interactions are being accounted for in the estimated effects.

All the regressions presented here are multivariable, and included all the listed covariates. We have specified this explicitly in the Results section (L679-680).

6. While less essential, the reviewers made several other suggestions/queries that should be considered by the authors in their revision.

In response to points from the individual reviews, we have changed the following:

– Table 3 (previously and erroneously 4) has been revised to cut down decimal places and include a separate column giving Wald *p*-values for the inclusion of each variable.

– All the supplementary regression tables (S4-S7) have been adjusted in the same way, although we have kept the greater precision in estimates in the supplement.

– The regression models were also changed such that pregnancy included as an interaction term with sex

– Reviewer 1 asked for model predictions in table 3. We have calculated these for a set of hypothetical patients with typical comorbidities, but present it as a separate figure (new figure 3) instead.

– We agree that it is possible that some patients that we previously classified as having nosocomial infections did, in fact, get infected in the community and were subsequently admitted for a different medical condition before COVID-19 symptoms appeared. As such, we have changed how this is represented in the text (L87-89).

– We have included some discussion and references on the apparently protective effect of asthma (L1053-1059).

– Our interpretation of the low ICU/HDU admission rates in the elderly is that they are the result of clinical judgment and/or patient preference (L1003-1005). This would also apply to comorbidities, which had effects that did not disappear once age was controlled for; we have added words in this discussion to reflect this (L1005-1006).

– The text “Patients lost to follow-up before any of these outcomes were included unless the time to that event was the outcome of interest” merely indicated that, for example, a patient lost to follow-up before final outcome would not be included in the analysis of time to outcome, but *would* be included in the time to ICU analysis if that information was available. This has been clarified in the text (L95-97).

– We have added an additional discussion of the difficulty in establishing causality, using the reduced odds of ICU/HDU admission in patients admitted before symptom onset as an example. (L1055-1063)

– The statement “patients admitted after experiencing symptoms for longer than 1 week were less likely to die” is justified by our results, as the CFR for death was lower in that group (see table 3). We have, however, clarified this text somewhat, as it was intended as part of our explanation as to how the relationship between longer time to admission and better outcomes might work (L968-970).

– The number of patients included in each analysis, and the proportion this forms of the total, are included at the start of each section.

Reviewer #1:[…] 1. Please provide date of end of follow-up and an estimate of the average/total follow-up time (STROBE item 14C).

These are now provided (see comments to editor above).

2. Please provide more information regarding participant recruitment, such as numbers potentially eligible and recruited into the study (STROBE item 5,6,13). Were all eligible COVID-19 cases included, or were only some included? Are there more eligibility criteria than those mentioned? The discussion suggests it is the latter (which the authors refer to as "enrolment bias", would recommend rephrasing this as selection bias). If site case loads influenced recruitment then this could potentially lead to selection bias if certain types of patients are favored for study inclusion and this varies over time.

The decision to enrol patients was taken by sites according to their capacity to do so at the time. We are not, unfortunately, privy to those decisions. There is likely some degree of selection bias involved at the individual site level, but it is not clear that this would behave in a consistent manner between different sites. Furthermore, all enrolments were done prospectively and staff would be blind to our outcomes of interest. See lines 955-964 for expanded discussion of these issues.

3. Please report in the main text the number (%) of patients lost to follow-up or with missing data for each analysis (STROBE item 13,14).

These are now provided (see comments to editor above).

4. It is unusual to have an outcome of odds or probability of an event where the time frame over which the patient can experience the outcome is not specified. Presumably the outcomes for question 2 and 4 are ICU admission or death over the entire study period. This is not a fair comparison, because patients recruited in later months may not have been followed-up as long as patients recruited earlier, and the follow-up of patients may have varied over the duration of the study. It is generally more rigorous to specify a time frame over which the outcome is assessed and only analyze patients with that follow-up time (ex. death within 3 months of admission) or to use analysis methods which account for differential follow-up time of participants, such as Kaplan-Meier or survival analyses. I would suggest modifying the logistic regressions to have the outcome over a specified time frame or to use survival analyses instead (or to explain why these methods were not used).

See comments to the editor above.

5. P39: "Patients lost to follow-up before any of these outcomes were included unless the time to that event was the outcome of interest" It is unclear to me what this means. Does this mean that patients lost to follow-up are included as non-events in the denominator?? It seems to me a better way to treat these patients would be to exclude them or better yet to include them as right-censored observations in survival analyses, as suggested above.

Patients lost to follow-up were indeed excluded; this text has been clarified to indicate what was intended (L95-97).

6. Table 4:– There is an excessive number of decimal places in this table (pseudo-precision) that are not clinically meaningful. I would suggest rounding all values to the nearest decimal (odds ratios) or to the nearest percentage point (percentages). Reducing the number of decimal points will make this table much simpler to read and understandable.

We have reduced this table to one decimal place for the main text version. The supplementary regression tables retain more precision.

– Please consider dropping the p-values for each individual value. The confidence intervals are much more informative as they indicate the precision around each estimate, convey the same information, and put the emphasis on the estimation of the parameter rather than its statistical significance. The small p-values in most cases are a reflection of the large sample size.

We agree and have removed all individual coefficient *p*-values.

– Conversely, the addition of p-values for the Wald type III test for the overall effect of each variable to the table would be informative.

These have been added.

– The above suggestions (decreasing decimal places and dropping p-values) should create more space in the table. Please consider using this space to add descriptive data on the probability of experiencing death and the number days to death or discharge for each category. This descriptive information helps to contextualize the odds ratios and relative time increases, and may be more clinically useful for some readers than the ratio measures.

We have chosen to add this information as a new figure instead (figure 3).

– It is not clear whether the results from this table present univariate or multivariable regression model results (presumably the latter). If the analyses are multivariable, then all the variables that were included in the model should be described in a footnote, as this influences the interpretation of the results. For example, it is important to know whether the monthly estimates are adjusted for sex and age, as the age and sex of patients varied over time.

All regression results are multivariable; this has been explicitly specified (L689-690).

– The pregnancy variable is conflated with the effect of sex because presumably all the men where categorized as 0 for this variable. This variable should be treated as an interaction variable with sex (sex + sex*pregnancy) in order to separate the effect of sex from the effect of pregnancy.

This adjustment to the relevant regression models has been done.

7. There is no Table 3.

We apologise for this typo; it has been corrected.

8. The authors mention system capacity during epidemic surges as a potential contributor to time trends, but did not include any indicator of system capacity in their analysis, so this interpretation remains speculative. Have the authors considered included such indicators as a variable in their multivariable model (ex. number of COVID-19 infections or hospitalizations reported during the day/week of admission in each country)?

Unfortunately, we did not have access to information of this sort on a level comprehensive enough to use it. We have acknowledged this as a limitation (L1087-1089).

Reviewer #2:[…] Although the paper does have strengths in principle, the weaknesses of the paper are that these strengths are not directly demonstrated. In particular:1. In this study, why gastrointestinal symptoms (abdominal pain, diarrhoea, and vomiting) are emphasized?

We have chosen to remove these symptoms as variables in all the regression models (see comments to editor above).

2. In this study, patients with a recorded hospital admission date before their symptom onset date, were taken to be a nosocomial infection. But patients who had got infection after exposure to virus may present symptoms later after admission. That is ,on the day of admission, the patient may be in the incubation period.

We agree that a minority of these individuals were probably not genuine nosocomial infections, and have changed how we refer to them throughout.

3. Similarly, "patients with nosocomial infections had lower odds of being admitted to ICU/HDU (OR 0.68, 95% CI 0.62-255 0.74)". The underlying reason may be not nosocomial infection, but that these patients had been admitted earlier than those with adverse outcomes.

We concur that this is not necessarily a result of a difference in the severity of nosocomial infections, and that causality is difficult to establish; we now discuss this example (L1074-1082).

4. Line 205: "times" should be "time". Similarly, in the lines 209 and 218.

This text has been adjusted.

5. For admission to ICU as an example, the comparison between different months means little, as there are many confounding factors for admission to ICU. One and half years have passed, it is time to look forward to improve treatment.

We agree that using numbers calculated from some time in the past is of limited use in guiding future decisions. Indeed, this is one of our key arguments; that conditions do change during a pandemic and this needs to be accounted for in planning and clinical trials.

6. Why less elderly patients were admitted to ICU?

We hypothesise that this was due to both clinical prognosis and patient preference (L1022-1024)

7. Why a wide variety of serious or chronic medical conditions were associated with lower odds of ICU admission. Is it because of lower admission of elderly patients?

These effects persisted once age was accounted for, but we suspect the reasons were similar. We have now stated this explicitly (L1024-1025).

8. The raw case fatality rate (CFR) was extremely high in this paper. The readers would be interested in the representativeness of this cohort.

See the discussion in the comments to the editor.

9. In this study, the authors confirmed what is known that Age, male and comorbidities are risk for higher mortality. The readers would like to understand why asthma was associated with lower mortality.

We now discuss the asthma findings in the context of the literature (L1053-1059).

10. Discussion: "patients admitted after experiencing symptoms for longer than 1 week were less likely to die." I am afraid that the explanation lacks scientific evidence.

We did find lower CFRs in patients waiting for a longer period before admission; we suspect this is due to differences in patient behaviour in response to their symptoms. We do, however, accept that the intended meaning of this passage was not entirely clear, and have reworded it (L987-989).

[Editors' note: further revisions were suggested prior to acceptance, as described below.]

The manuscript has been improved but there are some remaining issues that need to be addressed, as outlined below:1. The authors mention that the time periods over which the probability of ICU admission and death/discharge were measured are implicit in the cutoff times for time to event analyses (13 days and 45 days). This should be made explicit in the manuscript by indicating that the outcomes were ICU admission within 13 days for question 2 and death/discharge within 45 days for question 4.2. While the authors specify that all analyses are multivariable in the text, this should also be mentioned in the table legends and/or footnotes in order for tables to be self-sufficient.3. Table 1: because 'nosocomial infection' is no longer defined in the methods, this row should be relabeled as post-admission symptom onset infections4. There is an error in Table 2, the rows for symptoms at admission are mistakenly labeled as comorbidities and vice-versa.5. The authors refer to an non-existent supplementary figure 2 in the results. Presumably they mean Figure 2. This should be fixed.6. Figure 4 is missing a C inset.7. On line 446 the authors refer to Figure 3 when they should be referring to Figure 4.8. Table 3 appears to be lacking the standard deviation for time from symptom onset to hospital admission.

All the requested changes have been made, with the exception that, regarding point 8, the problem was in fact simply a mis-aligned column header.

In addition, in responding to point 1, we realised that we also needed to make a change to the results of question 2. In order for the variable of interest to be ICU/HDU admission within 13 days, we actually needed to exclude patients with less than 13 days of follow-up, no ICU/HDU admission, and unknown outcome. Figure 2A and the relevant regression table have been changed accordingly, with minimal effect on the results.